# Single-cell expression profiling of bat wing development

Xue Lyu [1,4], Jing Bai[1,2,4], Ji-Bin Jiang[1,2,4], Chang-Jie Sun[1,2], Peng Chen[1], Qi Liu [1], Yuan-Shuo Ma[1,2] & Zhen Liu [1,3] ✉

Bats are the only true-flight mammals, with wings formed by elongated digits and wing membranes. Despite the uniqueness, the cellular and molecular aspects of bat wing development remain largely unknown. Here, we use single-cell transcriptomic sequencing to map ~39,000 cells from the limbs of bats (*Rhinolophus sinicus*) at developmental stages Carnegie stages (CS) 16, 18, and 20. We identify 16 distinct cell populations, including a specific mesenchymal progenitor population (*PDGFD*+) in bat forelimbs, which may differentiate into the interdigital membrane and promote bone cell proliferation. Developing bat forelimbs exhibit prolonged chondrogenesis and delayed osteogenesis, resulting in more chondrocytes and fewer osteoblasts. The integrative analyses of data from single-cell and bulk RNA sequencing highlight the crucial roles of Notch signaling activation and WNT/β-catenin signaling suppression in bat forelimb development. Our findings provide a comprehensive single-cell atlas of developing bat limbs, offering insights into the mechanisms underlying bat wing development.

Bats are the only mammals capable of truly self-powered flight, an exceptional evolutionary adaptation that has made them one of the most adaptable mammalian species[1]. The evolution of powered flight in bats can be traced back to ~56 million years ago, as evidenced by fossil findings showcasing anatomical traits like elongated digits and flight membranes[2]. These distinctive evolutionary characteristics of bat forelimbs have sparked research interest in unraveling their molecular and developmental mechanisms. For instance, histological comparisons have shown that the developing bat forelimbs have larger areas of chondrocyte proliferation than mouse forelimbs[3]. Bats maintain the interdigital membranes of their forelimbs by combining increased fibroblast growth factors (FGFs) and decreased bone morphogenetic protein (BMP) signaling[4]. Transcriptomic analyses of developing bat limbs have identified several genes, such as *MEIS2*, *TBX3*, *BMP3*, *FGF8*, and *Hox*, that are upregulated in the forelimbs compared to the hindlimbs[5,6]. These findings have greatly improved our understanding of bat forelimb development.

Nevertheless, it is important to note that the expansion of interdigital membranes occurs concurrently with digit elongation during the development of bat forelimbs. This developmental coordination among local tissues is often orchestrated by interactions among cell progenitors, which influence the diversity and transitions in cell states during development[7]. The challenge in distinguishing between digits and interdigital membranes in bat forelimb development has led to a limited understanding of how digit elongation and membrane expansion are coordinated developmentally.

In this study, we addressed this issue by conducting an in-depth single-cell transcriptomic analysis of developing bat limbs. Our systematic investigation uncovered the cellular and molecular factors contributing to the specialization of bat forelimbs by comparing them with bat hindlimbs and mouse forelimbs. Our data suggested that interactions facilitated by growth factors among skin- and bone-related cell populations play a crucial role in coordinating the development of elongated digits and interdigital membranes in bats.

[1]State Key Laboratory of Genetic Evolution & Animal Models, Kunming Institute of Zoology, Chinese Academy of Sciences, Kunming, China. [2]University of Chinese Academy of Sciences, Beijing, China. [3]Yunnan Key Laboratory of Biodiversity Information, Kunming, China. [4]These authors contributed equally: Xue Lyu, Jing Bai, Ji-Bin Jiang. ✉e-mail: zhenliu@mail.kiz.ac.cn

## Results

### Key developmental stages of bat forelimbs

While there have been detailed descriptions of bat development, such as studies focusing on the short-tailed fruit bat (*Carollia perspicillata*)[8] and the Natal long-fingered bat (*Miniopterus natalensis*)[9], a comprehensive understanding of the critical developmental stages of bat forelimbs remains incomplete. This is due to the lack of systematic quantification using developmentally stable structures as controls and the absence of cross-species comparisons. To address this question, we conducted skeletal staining on embryonic forelimbs and hindlimbs of two bat species, the Chinese horseshoe bat (*Rhinolophus sinicus*, RSI) and the eastern bent-winged bat (*Miniopterus fuliginosus*, MFU), as well as laboratory mice (Fig. 1a). This analysis spanned consecutive developmental stages from Carnegie Stage 16 to 21 (CS16 to CS21), corresponding to mouse developmental stages E13.5 to E17.5[9].

We assessed the developmental dynamics of digits I–V by measuring the lengths of metacarpals and phalanges for the three species (Supplementary Data 1). To account for varying developmental rates across different body regions and species, we used cranial width as a control, as it exhibits relatively consistent developmental rates across diverse mammalian species[10]. Our analysis showed that the development of the short digits I and II in the forelimbs of both bat species remained relatively consistent compared to their hindlimbs and mouse limbs across various developmental stages (Supplementary Fig. 1). In contrast, the developmental rate of the elongated digits III, IV, and V in the forelimbs of the two bat species did not show differences at CS16, started to diverge at CS18, and exhibited significant differences at CS20 compared to their hindlimbs and mouse limbs (Fig. 1b and Supplementary Fig. 1). Notably, the interdigital membranes in the developing bat forelimbs were not only retained during early stages (CS16 and CS17) but also expanded progressively alongside digit

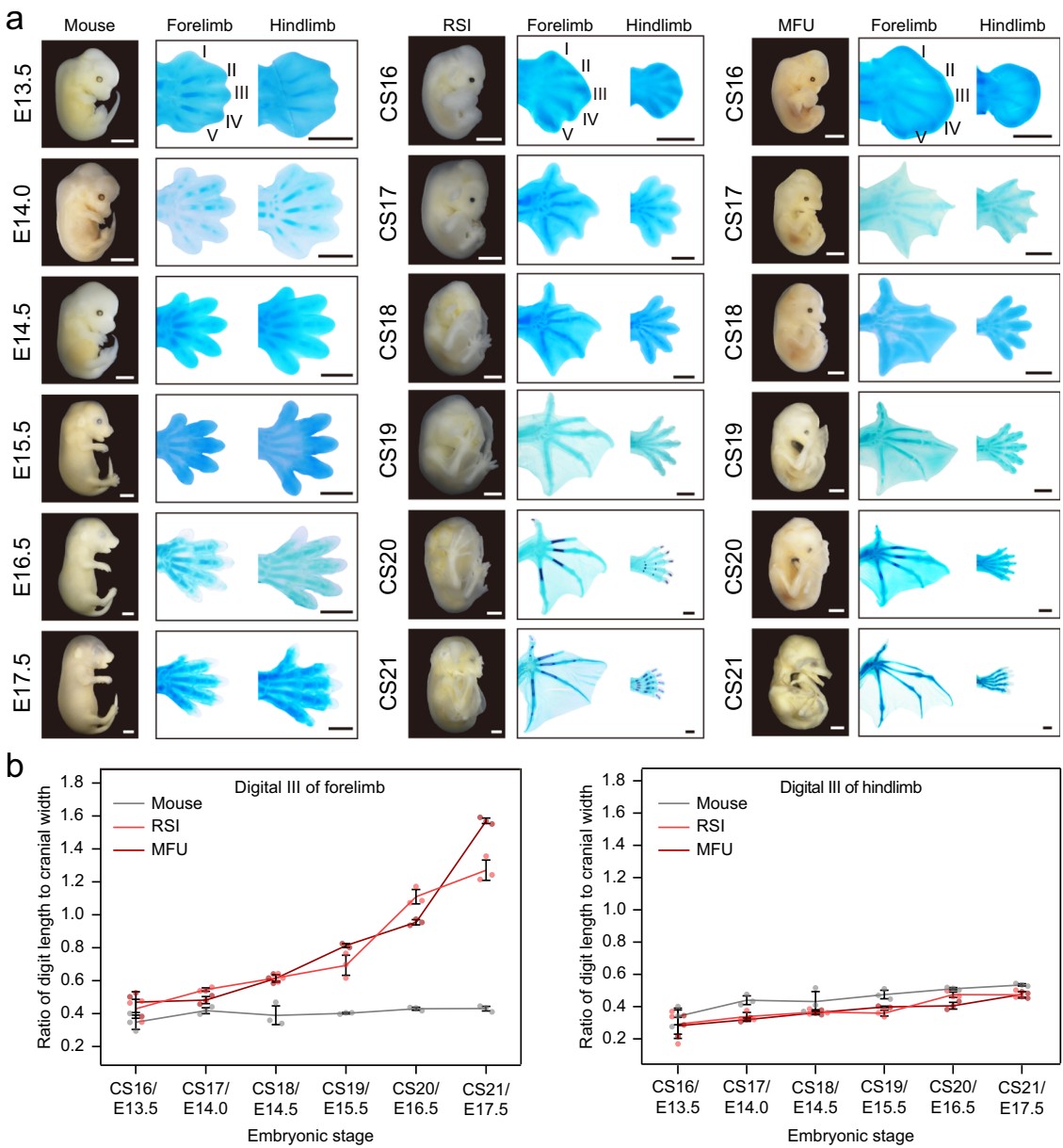

**Fig. 1 | Developmental dynamics of bat limbs. a** Embryos of the laboratory mouse, Chinese horseshoe bat (*Rhinolophus sinicus*, RSI), and eastern bent-winged bat (*Miniopterus fuliginosus*, MFU) with alcian blue staining of forelimbs and hindlimbs across six developmental stages. Scale bars: 2 mm for embryonic images and 1 mm for alcian blue staining. **b** Ratios of digit III length to cranial width across six developmental stages for the forelimbs and hindlimbs of the two bat species and laboratory mice. Data are presented as mean values ± SD. The numbers of dots represent the numbers of independent biological replicates.

elongation at the later developmental stages from CS18 to CS21 (Fig. 1a). These findings indicate notable developmental heterochrony between bat forelimbs and hindlimbs compared to mouse forelimbs and hindlimbs, suggesting that CS16, CS18, and CS20 are pivotal developmental stages for digit elongation and interdigital membrane expansion in bat forelimbs.

## A single-cell census of the developing bat limbs

To explore the cellular and molecular characteristics underlying the specialization of bat forelimbs, we generated single-nucleus transcriptomes from RSI forelimbs and hindlimbs at CS16, CS18, and CS20 using the SPLiT-seq method[11]. We sequenced 16 libraries on the Illumina NovaSeq 6000 platform, yielding 288.4 Gb of clean reads (Supplementary Data 2). After quality controls, we profiled 38,942 cells: 20,003 from forelimbs and 18,939 from hindlimbs based on single-cell combinatorial indexing (Supplementary Data 3). These cells were further classified into stages CS16 (forelimb 9891/hindlimb 9264), CS18 (4004/3317), and CS20 (6108/6358). We applied the uniform manifold approximation and projection (UMAP) method[12] for the dimensionality reduction and clustering of these single-cell transcriptomes (Fig. 2a), identifying 16 transcriptionally distinct cell populations (Fig. 2b). These populations were annotated using previously established marker genes[13–15] (Fig. 2c). Six populations were identified as mesenchymal progenitors (MPs), with specific marker gene expressions such as *PNISR*, *EBF2*, *ARHGAP24*, *MEIS2*, *PDGFD*, and *ZFHX3* (Fig. 2b). Additionally, we identified prominent cell populations like chondrocytes, chondrocyte precursors (CPs), osteoblasts, and osteoblast precursors (OPs) (Fig. 2b). To confirm the reliability of our clustering and annotations, we analyzed the top 2000 variable genes from each cell population to examine their expression correlations across different populations in the developing bat forelimbs and hindlimbs. This analysis showed that the highest expression correlation occurred between similar cell populations within either bat forelimbs or hindlimbs, as well as between identical cell populations in bat forelimbs and hindlimbs (Supplementary Fig. 2). Additionally, we identified highly expressed genes in each cell population to conduct tissue and functional enrichment analyses using the ENRICHR and Metascape databases, respectively[15,16]. The results showed that the enriched tissue types and functional categories closely aligned with our cell population annotations (Fig. 2d, e). These findings suggest a high level of reliability in our clustering and annotation of the cell populations involved in the development of bat limbs.

## More chondrocytes in developing bat forelimbs

Next, we compared the abundance of each of the 16 cell populations between developing bat forelimbs and hindlimbs (Fig. 3a). To ensure fair comparisons, we normalized the cell numbers for each population against the total number of cells from each tissue across different developmental stages. Statistical analysis revealed that 11 out of 16 cell populations (68.8%) had significant differences in cell proportions between bat forelimbs and hindlimbs (Fig. 3b), indicating a considerable variation in cell abundance across different cell populations in the developing bat forelimbs and hindlimbs. Specifically, the proportion of chondrocytes was significantly higher in developing bat forelimbs (10.5%) compared to hindlimbs (6.4%) ($P < 0.0001$, $\chi^2$ test), while the proportion of osteoblasts was significantly lower in developing bat forelimbs (2.5%) than in hindlimbs (4.8%) ($P < 0.0001$, $\chi^2$ test; Fig. 3b).

To explore the developmental dynamics of chondrocytes and osteoblasts during the development of bat limbs, we classified the two cell populations into different developmental stages (Fig. 3c). While the numbers of both chondrocytes and osteoblasts showed a gradual increase, the proportion of chondrocytes exhibited a more rapid rise from CS16 to CS18 in bat forelimbs compared to hindlimbs (Fig. 3d). In contrast, the proportion of osteoblasts increased more rapidly from CS16 to CS20 in bat hindlimbs compared to forelimbs (Fig. 3d). These results suggest that the continuous rapid proliferation of chondrocytes and slow ossification may contribute to the digit elongation during the development of bat forelimbs.

## Specific *PDGFD*⁺ MPs in developing bat forelimbs

In addition to the developmental differences of chondrocytes and osteoblasts between bat forelimbs and hindlimbs, we identified two intriguing populations of MPs that highly expressed the Meis Homeobox 2 gene (*MEIS2*) and the platelet-derived growth factor D gene (*PDGFD*), respectively (Fig. 4a, b). The proportions of these two cell populations were significantly higher in bat forelimbs (7.2% and 11.5%) compared to hindlimbs (0.9% and 0.7%) when normalized against the total number of cells within each limb across all stages ($P < 0.0001$, $P < 0.0001$, $\chi^2$ tests; Fig. 4c). In terms of developmental dynamics, the proportion of *MEIS2*⁺ mesenchymal progenitors (MMPs) was ~11.7% at CS16, gradually decreasing to ~0.3% at CS20 in bat forelimbs, while the proportion of MMPs remained very low in bat hindlimbs (CS16: 1.7%; CS18: 0.4%; CS20: 0.05%; Fig. 4d). The results suggest that MMPs may represent a forelimb-specific, temporal cell population that differentiates into other cell states in later stages of bat development. Conversely, the proportion of *PDGFD*⁺ mesenchymal progenitors (PDMPs) was ~1.1% at CS16 and increased to ~21.8% at CS20 in bat forelimbs, while remaining low in the developing bat hindlimbs (CS16: 0.03%; CS18: 0.8%; CS20: 1.7%; Fig. 4d). These findings suggest a crucial role of PDMPs in the development of bat forelimbs.

To confirm the specificity of MMPs and PDMPs in the developing bat forelimbs, we conducted single-cell transcriptome sequencing for mouse forelimbs at E13.5, E14.5, and E16.5 (Supplementary Data 2), corresponding to bat stages CS16, CS18, and CS20[9]. We profiled 34,172 high-quality cells from mouse forelimbs: 4818 at E13.5, 19,632 at E14.5, and 9722 at E16.5. Comparing these cells with the annotated bat dataset revealed that very few cells were closely associated with MMPs and PDMPs (0.4% and 0.006%; Supplementary Fig. 3). Additionally, we also retrieved and analyzed single-cell transcriptomic data from developing mouse hindlimbs at equivalent stages[17,18]. Mapping these cells onto the mouse forelimb and bat limb atlas confirmed that few cells were associated with MMPs and PDMPs in developing mouse hindlimbs (Supplementary Fig. 3), supporting the specificity of these progenitors in developing bat forelimbs.

To further validate these results, single-nucleus transcriptomes from RSI forelimbs and hindlimbs at CS21 were generated using an independent droplet-based scRNA-seq method (10× Genomics) (Supplementary Data 4). Mapping these cells onto the annotated cell topology (Supplementary Fig. 4a) revealed that the proportion of PDMPs was significantly higher in bat forelimbs (18%) compared to hindlimbs (2.8%; $P < 0.0001$, $\chi^2$ test) and nearly no MMPs were identified in bat forelimbs and hindlimbs at CS21 (Supplementary Fig. 4b). These findings align closely with observations from earlier developmental stages of bat limbs, highlighting the consistency and reliability of our results across different species and single-cell sequencing platforms.

Given the specificity of MMPs and PDMPs in developing bat forelimbs, we hypothesized that these cells might differentiate into the flight membranes of bat forelimbs. To test this, we conducted whole-mount in situ hybridization (WISH) using marker genes for MMPs and PDMPs. The results showed prominent expressions of *MEIS2* and *PDGFD* in the interdigital membranes of bat forelimbs at CS16, CS18, and CS20, with no expression detected in hindlimbs (Fig. 4e). To confirm the results, we used *SYNPO2* and *FREM1*, highly expressed in MMPs, and *CSRNP3*, highly expressed in PDMPs, as additional markers for RNA in situ hybridization. These analyses yielded highly consistent results (Supplementary Fig. 5), suggesting that MMPs and PDMPs are specific to developing bat forelimbs compared to hindlimbs, likely differentiating into interdigital membranes in the bat forelimbs. Notably, these marker genes generally showed higher expression in interdigital membranes between digits III, IV, and V, which were the

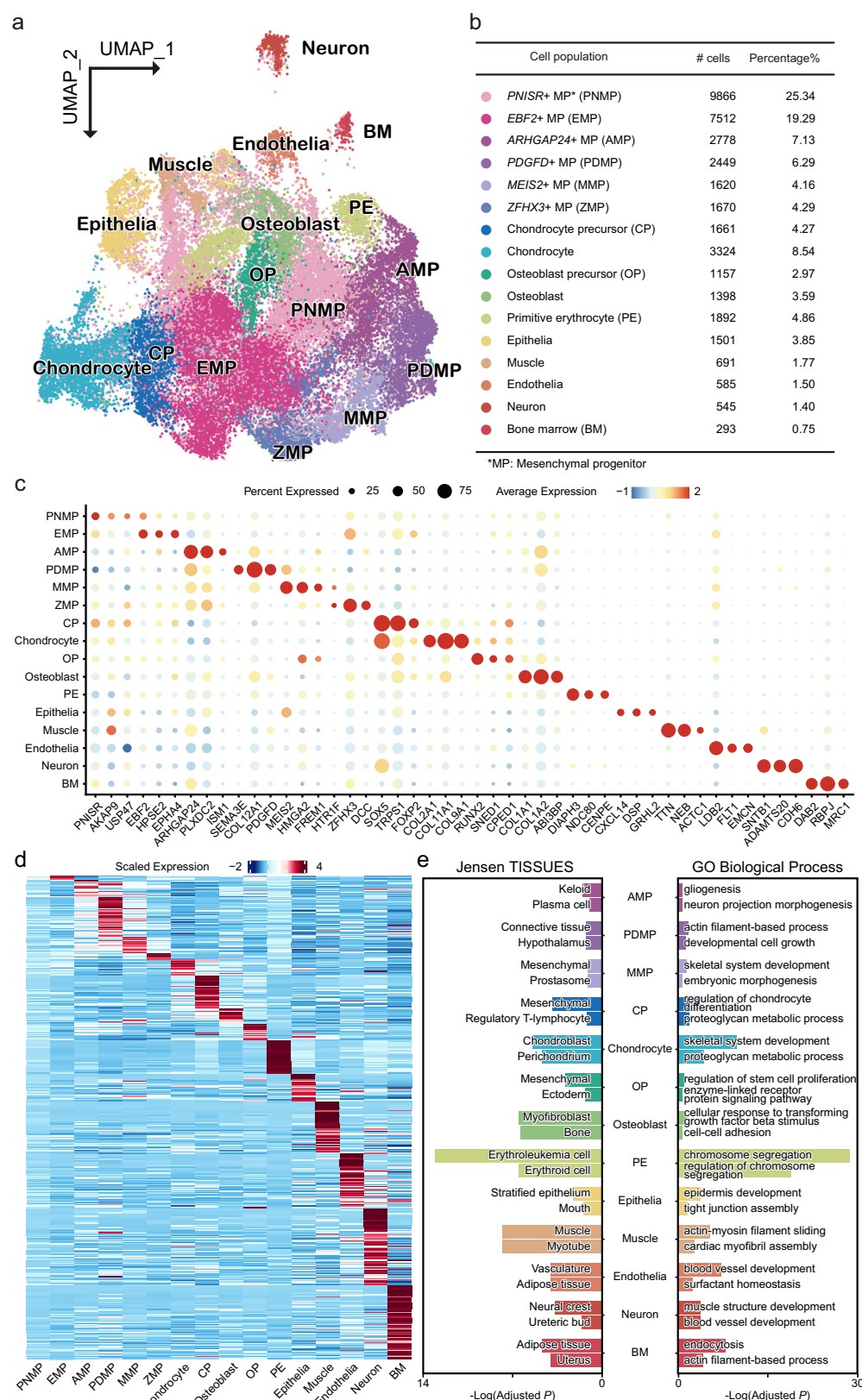

**Fig. 2 | Single-cell analysis of developing bat limbs. a** Autopods from *R. sinicus* forelimbs and hindlimbs at developmental stages CS16, CS18, and CS20 are analyzed by single-cell RNA sequencing. UMAP visualization of color-coded cell populations in developing bat forelimbs and hindlimbs. **b** Absolute cell numbers and proportions for each cell population. **c** Dot plot showing marker gene expression for each cell population. **d** Genes with specific high expression for each cell population. **e** Top two terms from the tissue and functional enrichment analyses for the highly expressed genes of different cell populations. The *P* values are from two-sided Fisher's exact tests and adjusted with the Benjamini–Hochberg method.

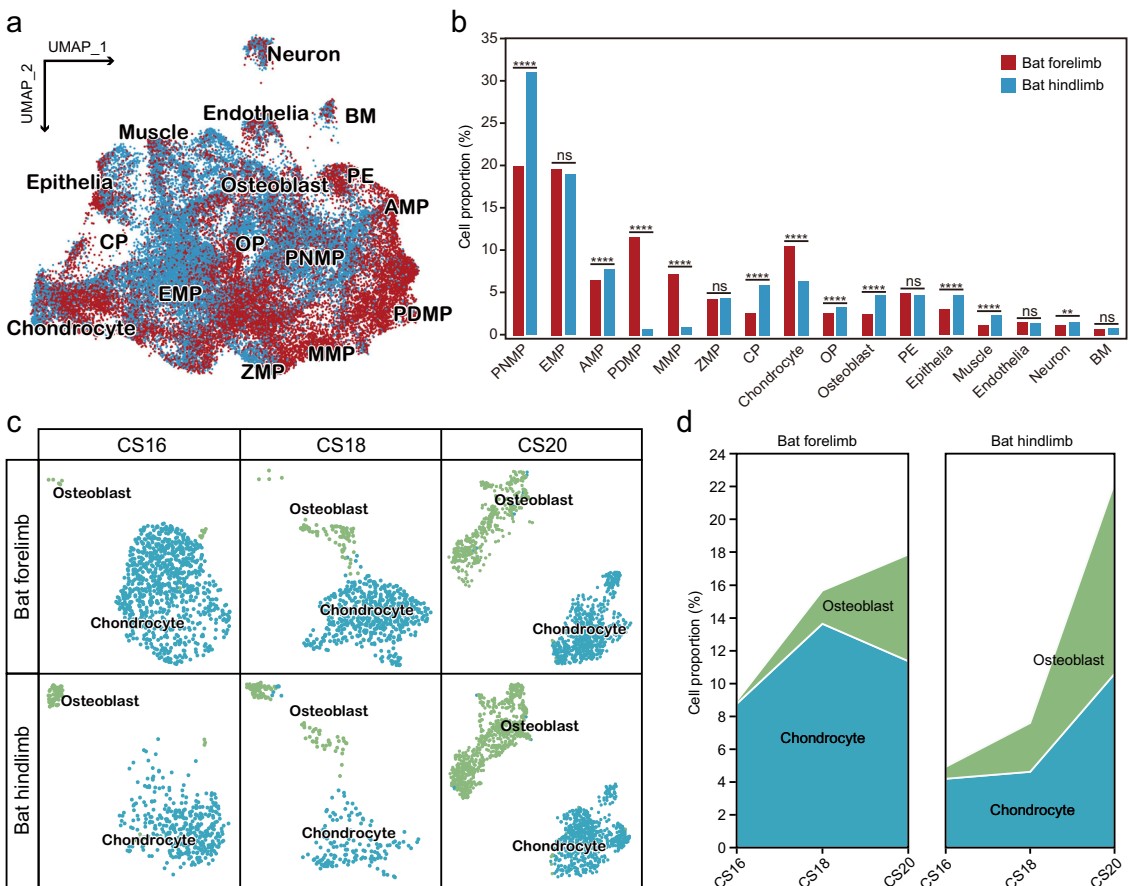

**Fig. 3 | Cell proportion differences across distinct cell populations between developing bat forelimbs and hindlimbs. a** UMAP visualization of various cell populations during the development of bat forelimbs and hindlimbs. **b** Comparison of cell proportions across various cell populations between developing bat fore-limbs and hindlimbs. **c** Developmental dynamics of chondrocytes and osteoblasts in bat forelimbs and hindlimbs across three developmental stages. **d** Changes in cell proportions of chondrocytes and osteoblasts between bat forelimbs and hin-dlimbs at three developmental stages. Statistical significance was determined using two-sided $\chi^2$ tests. *$P < 0.05$; **$P < 0.01$; ***$P < 0.001$; ****$P < 0.0001$; ns not significant.

most elongated digits, rather than between digits I, II, and III, sug-gesting an antero-posterior pattern during the development of bat interdigital membranes.

**Prolonged chondrogenesis in developing bat forelimbs**
To investigate the developmental origins of chondrocytes, osteoblasts, MMPs, and PDMPs, we included these cell populations along with other MPs to infer differentiation trajectories and pseudotemporal relation-ships using pseudotime trajectory inference and RNA velocity methods[19,20]. The analyses showed that *PNISR*[+] mesenchymal progeni-tors (PNMPs) were dispersed throughout all trajectories (Supplemen-tary Fig. 6), suggesting that PNMPs represent a primitive cell population with a diverse transcriptomic profile[21]. This interpretation was further supported by their generally high transcriptome-wide similarity to other cell populations (Supplementary Fig. 2). After excluding PNMPs, both methods revealed a distinct trifurcating path, in which *EBF2*[+] MPs (EMPs) diverged into chondrocytes, osteoblasts, MMPs, and PDMPs in developing bat forelimbs (Fig. 5a; Supplementary Fig. 7a). The inferred differentiation trajectories suggested that MMPs potentially serve as precursors for PDMPs. Similarly, EMPs were iden-tified as progenitors for mesenchymal and osteochondral cell popu-lations in developing bat hindlimbs (Supplementary Fig. 7b). The developmental dynamics of these cell populations indicated that as the proportion of EMPs decreased from CS16 to CS18 and CS20, chon-drogenesis increased in bat forelimbs, whereas osteogenesis became more prominent in bat hindlimbs (Fig. 5b; Supplementary Fig. 7c).

To validate the increased chondrogenesis and delayed osteo-genesis in developing bat forelimbs compared to hindlimbs, we compared temporal gene expression dynamics along the develop-mental trajectory from EMPs to chondrocytes, aligning their devel-opmental dynamics between bat forelimbs and bat hindlimbs by matching pseudotime points of dynamic gene expression[22]. The analysis revealed accelerated gene expression in developing bat forelimbs compared to hindlimbs (Fig. 5c), suggesting a faster chon-drogenesis in developing bat forelimbs. Similarly, analyzing the tra-jectory from EMPs to osteoblasts showed a delayed gene expression in developing bat forelimbs relative to hindlimbs (Fig. 5d), suggesting delayed osteogenesis in developing bat forelimbs. To confirm these results, we compared the temporal expression patterns of chon-drogenesis and osteogenesis between bat and mouse forelimbs. Consistently, developing bat forelimbs exhibited accelerated chon-drogenesis but delayed osteogenesis compared to developing mouse forelimbs (Supplementary Fig. 8). To validate these findings from the single-cell sequencing, we examined stained embryonic forelimbs and hindlimbs of bats. No significant ossification differences were observed between forelimbs and hindlimbs during the early stages (CS16 to CS19). However, at CS20, calcified bones appeared in hin-dlimb phalanges, whereas no such calcification was detected in forelimb phalanges (Fig. 1; Supplementary Fig. 9). To confirm the differing proliferation rates of chondrocytes between developing bat forelimbs and hindlimbs, we examined embryonic samples at CS18 and CS20, treated with EdU via intraperitoneal injection to label cell

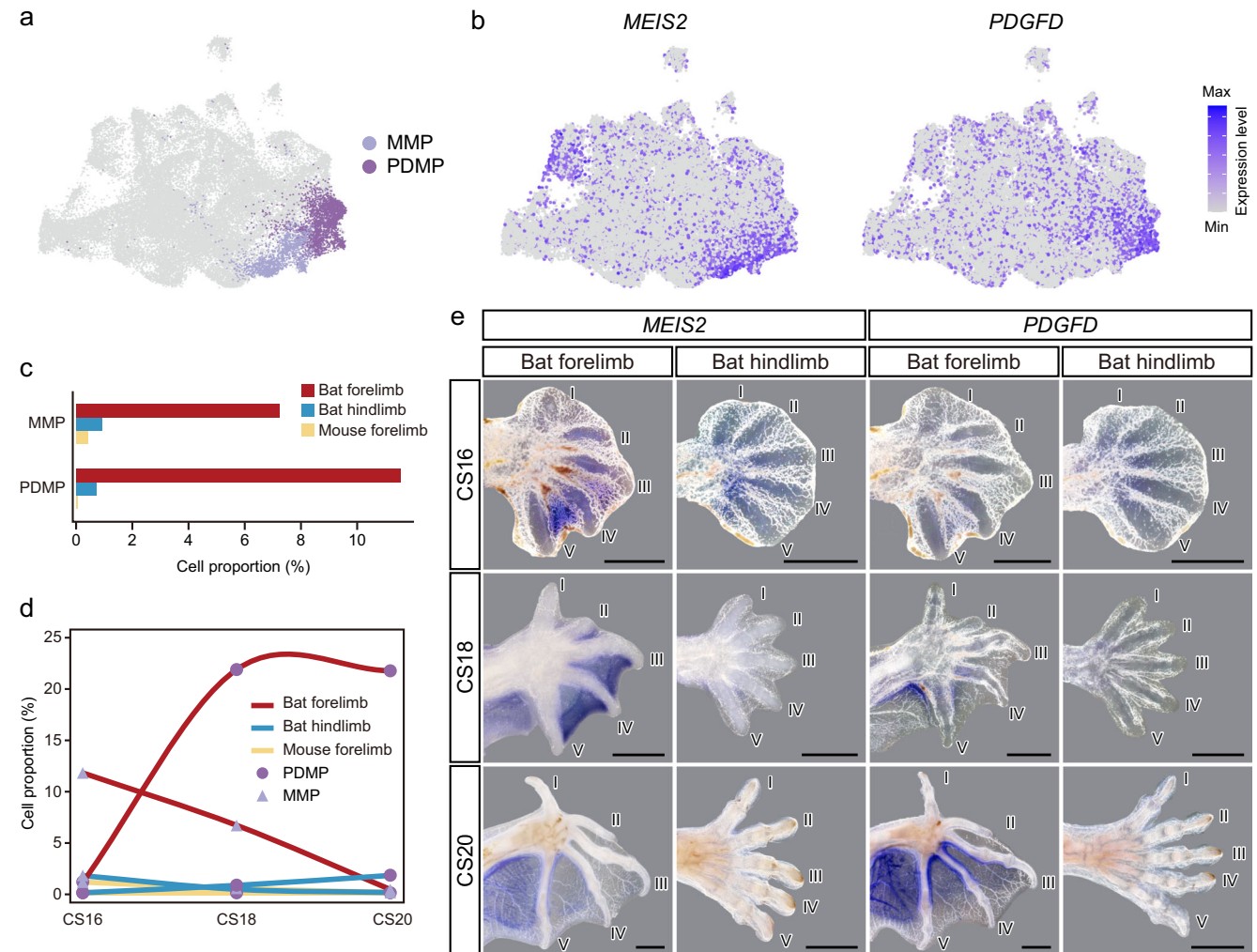

**Fig. 4 | Characterization of *MEIS2*⁺ and *PDGFD*⁺ mesenchymal progenitors (PDMPs and MMPs). a** UMAP visualization of PDMPs and MMPs. **b** Expression patterns of marker genes for MMPs and PDMPs. **c** Comparison of cell proportions for MMPs and PDMPs among bat forelimbs, bat hindlimbs, and mouse forelimbs. **d** Developmental dynamics of MMPs and PDMPs across three developmental stages in bat forelimbs, bat hindlimbs, and mouse forelimbs. **e** In situ hybridization for *MEIS2* and *PDGFD* in bat forelimbs and hindlimbs at stages CS16, CS18, and CS20. Scale bars: 1 mm.

proliferation. Using COL2A1 as a chondrocyte marker, we performed immunohistochemistry on the most elongated digit III (Fig. 5e). The results showed significantly higher EdU intensities in bat forelimb chondrocytes compared to hindlimbs at both stages ($P = 0.0024$, $P = 0.015$, two-tailed Student's *t*-tests; Fig. 5f). Additionally, to investigate differences in osteogenesis between developing bat forelimbs and hindlimbs, we selected ABI3BP as an osteoblast marker based on our scRNA-sequencing (Fig. 5g). The results indicated a significant increase in ABI3BP-positive cells in developing bat hindlimbs at CS20 compared to forelimbs ($P = 0.040$, two-tailed Student's *t*-test; Fig. 5h). These findings provide strong evidence of enhanced cell proliferation and delayed osteogenesis in developing bat forelimbs compared to hindlimbs. Interestingly, delayed ossification was widely observed in the forelimbs of newborn bats[23], supporting the reliability of our single-cell transcriptomic and experimental analyses of developing bat limbs.

Together, our findings suggest that *EBF2*⁺ MPs gradually transition into crucial cell populations, such as chondrocytes, osteoblasts, MMPs, and PDMPs during the development of bat limbs, In contrast to bat hindlimbs, the developing bat forelimbs demonstrate a prolonged developmental phase of chondrogenesis and a delayed onset of ossification, which may play a crucial role in the elongation of digits during the development of bat forelimbs.

## PTN and PDGF signaling pathways promote cell proliferation

Next, we conducted a quantitative analysis of intercellular communication networks using CellChat[7] to identify signaling pathways crucial for the specialization of bat forelimbs. In comparing bat hindlimbs and mouse forelimbs, we found that certain signaling pathways were upregulated in various cell populations within the developing bat forelimbs (Fig. 6a; Supplementary Figs. 10 and 11). Notably, the PTN (pleiotrophin) signaling pathway was significantly overrepresented in the developing bat forelimbs compared to both bat hindlimbs and mouse forelimbs ($P = 0.025$, $P = 1.86E-09$; paired Wilcoxon test; Fig. 6b). Network centrality analysis revealed that bone-related cell populations, including osteochondral progenitor EMPs, CPs, chondrocytes, and osteoblasts, were the primary sources and targets of the PTN ligand in developing bat forelimbs (Fig. 6c). The expression of *PTN* and its receptor *SDC2* was generally upregulated in these bone-related cells (Fig. 6d). These results suggest that PTN, a secreted heparin-binding growth factor, plays a critical role in enhancing cell proliferation, particularly in bone-related cells, in developing bat forelimbs. To confirm this, we created a HEK293T cell line that stably overexpressed PTN using lentivirus transfection and puromycin selection (HKE293T-PTN). Then we performed a Transwell assay, placing HEK293T-PTN cells in the upper compartment, while bat (*R. sinicus*) embryonic fibroblasts, mouse preosteoblasts (MC3T3-E1), and cells

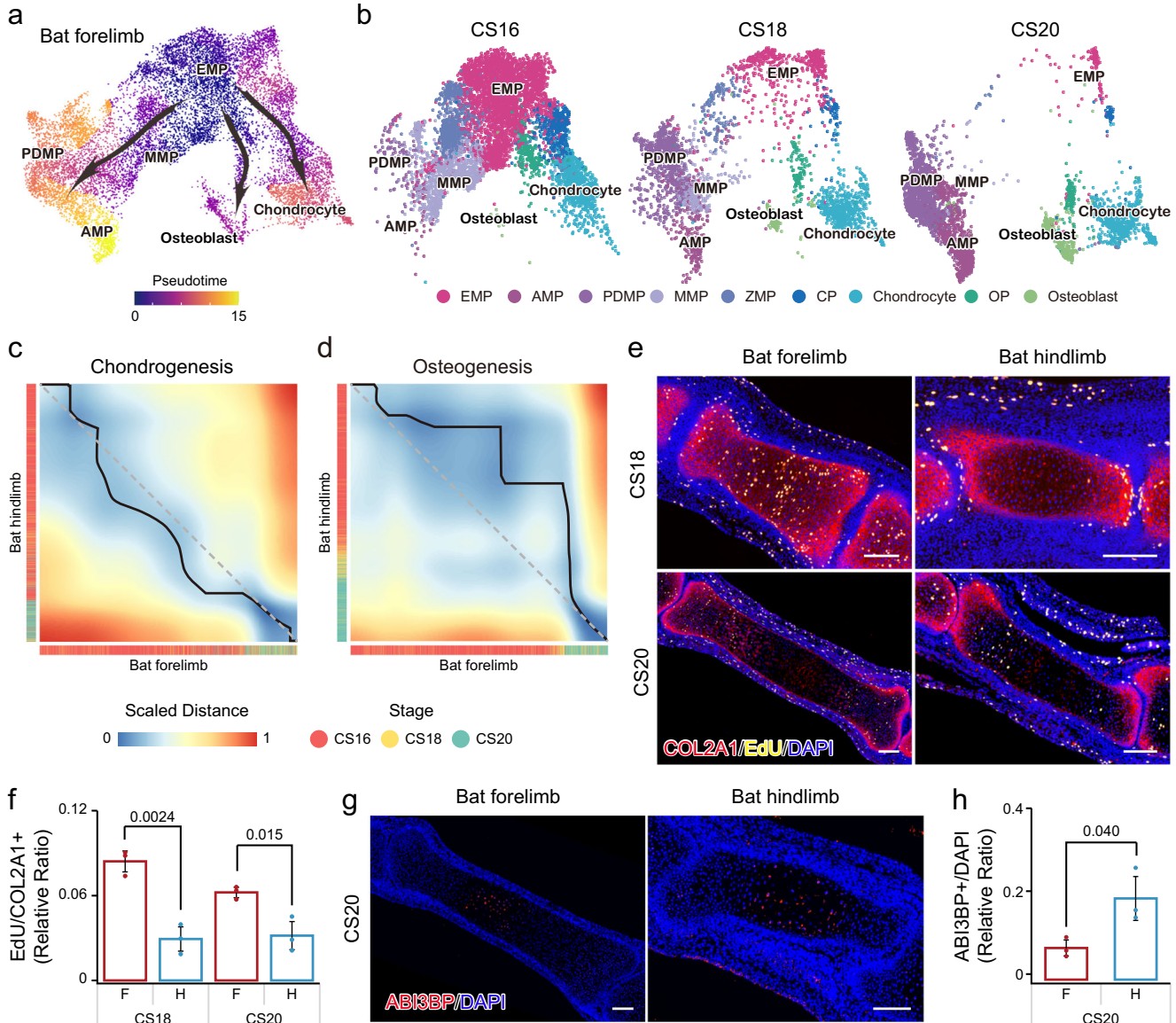

**Fig. 5 | Temporal dynamics of mesenchymal and osteochondral cell populations. a** Single-cell trajectories of mesenchymal and osteochondral cell populations in developing bat forelimbs, with arrows indicating predicted developmental paths. **b** Developmental dynamics of mesenchymal and osteochondral cell populations in bat forelimbs. Dissimilarity matrix, global alignment, and pseudotime shift from comparative alignment of gene modules along the trajectories of bat forelimbs and hindlimbs for chondrogenesis (**c**) and Osteogenesis (**d**). The histograms correspond to the developmental stages of bat limbs. **e** Immunofluorescent co-staining of COL2A1 and EdU labeling assay in the proximally first phalanx of digit III of bat

forelimbs and hindlimbs at CS18 and CS20, with DAPI for nuclei. Scale bar: 100 μm. **f** Relative ratio of EdU-positive cells to COL2A1-positive cells in developing bat forelimbs and hindlimbs. **g** Immunofluorescent co-staining of ABI3BP and DAPI in the proximally first phalanx of digit III at CS20. Scale bar: 100 μm. **h** Relative ratio of ABI3BP-positive cells to DAPI-positive cells in developing bat forelimbs and hindlimbs. The number of dots (n = 3) in (**f**, **h**) represents the number of independent biological replicates. All data are presented as mean ± SD. P values are from two-tailed Student's t-tests. Source data are provided as a Source Data file.

from bat embryonic forelimbs were placed in the lower compartment, respectively. After 72 h of co-culture, we observed a general increasing trend in the proliferation of bat embryonic fibroblasts, mouse pre-osteoblasts, and bat embryonic forelimb cells compared to control groups (P = 0.041, P = 0.046, P = 0.01; two-tailed Student's t-tests; Fig. 6e).

Due to the specificity of PDMPs, we examined the signaling interactions of this cell population with others in developing bat forelimbs. Among the enriched pathways, the PDGF signaling pathway was the most significantly represented in PDMPs (Fig. 6f). PDMPs predominantly served as the source of the PDGFC (Platelet-Derived Growth Factor C) ligand compared to other MPs, with CPs and OPs identified as primary recipients of this signal (Fig. 6g). The increased

expression of the *PDGFC* ligand in PDMPs and its *PDGFRA* receptor in CPs and OPs highlighted the strong connections between PDMPs and the precursors of chondrocytes and osteoblasts in developing bat forelimbs (Fig. 6g). These results suggest that PDMPs may contribute not only to interdigital membrane differentiation but also to the proliferation of other cell populations, including skin- and bone-related cells, by supplying growth factors during bat forelimb development. To validate this, we created a HEK293T cell line over-expressing *PDGFC* using the same strategy above and conducted a Transwell assay. HEK293T-PDGFC cells were placed in the upper compartment, while bat embryonic fibroblasts, mouse pre-osteoblasts, and bat embryonic forelimb cells were placed in the lower compartment, respectively. After 72 h of co-culture, PDGFC

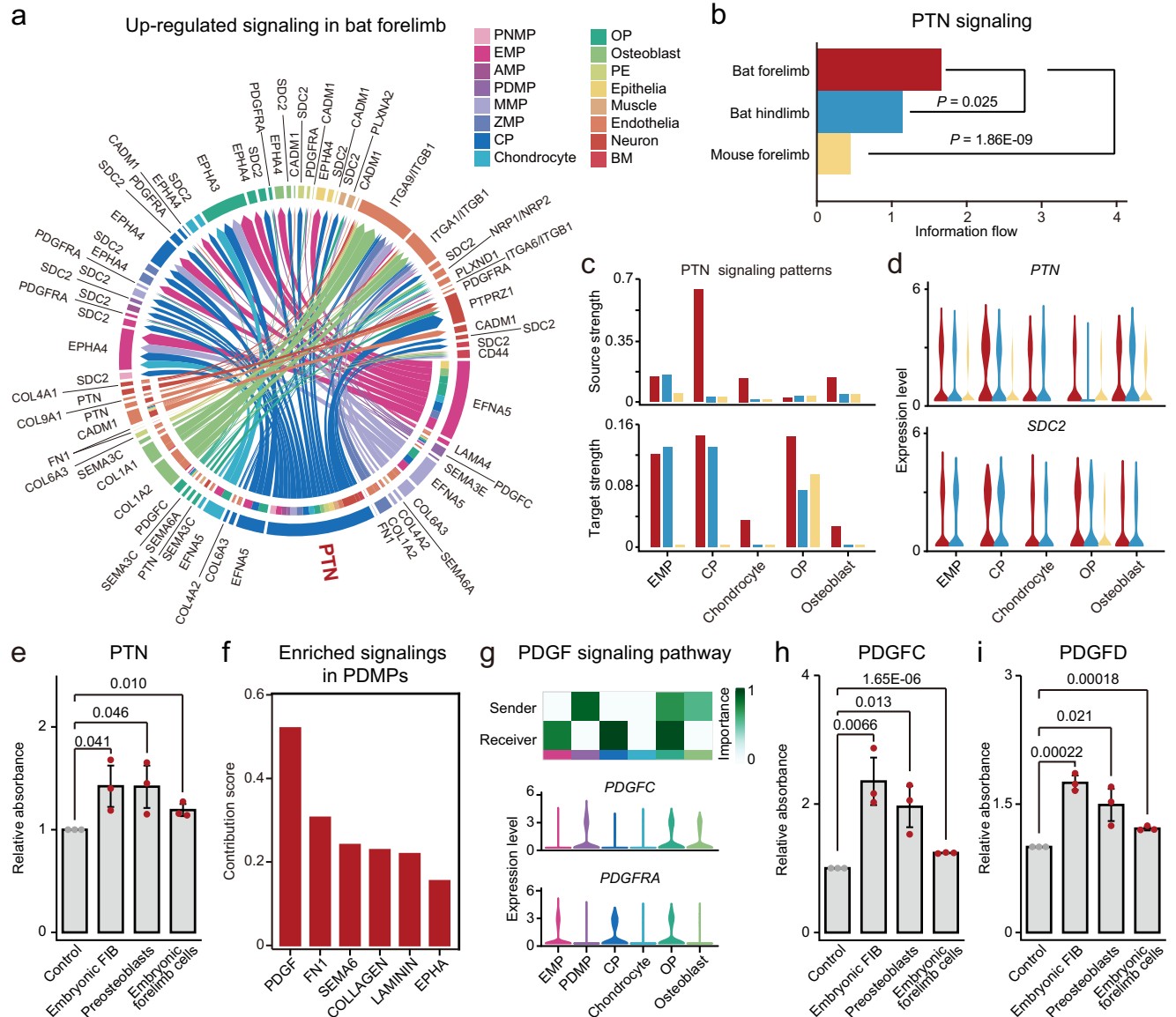

**Fig. 6 | Intercellular communications reveal key signaling events associated with the development of bat forelimbs. a** Overview of upregulated signaling pathway networks in developing bat forelimbs compared to hindlimbs. **b** Enhanced information flow in the PTN signaling pathway in the developing bat forelimbs compared to bat hindlimbs and mouse forelimbs. **c** Comparison of source and target strength for the PTN signaling pathway in EMPs and four osteochondral cell populations among bat forelimbs, bat hindlimbs, and mouse forelimbs. **d** Expression distribution of the *PTN* ligand and its *SDC2* receptor. **e** PTN significantly promotes the proliferation of bat embryonic forelimb cells, bat embryonic fibroblasts, and mouse preosteoblasts. **f** Enriched signaling pathways in PDMPs of developing bat forelimbs. **g** Relative contribution of each ligand-receptor pair in the PDGF signaling pathway and expression distribution of the *PDGFC* ligand and its *PDFGRA* receptor across osteochondral cell populations and their pre-cursors. PDGFC (**h**) and PDGFD (**i**) significantly promote the proliferation of bat embryonic forelimb cells, bat embryonic fibroblasts, and mouse preosteoblasts. The dots (*n* = 3) represent the number of independent experiments. All data are presented as mean ± SD. The *P* values are from two-tailed Student's *t*-tests. Source data are provided as a Source Data file.

significantly accelerated the proliferation of bat embryonic fibroblasts, mouse preosteoblasts, and bat embryonic forelimb cells (*P* = 0.0066, *P* = 0.013, *P* = 1.65E-06; two-tailed Student's *t*-tests; Fig. 6h). Further validating the roles of PDMPs, we repeated the Transwell assay for the marker gene *PDGFD* of PDMPs. The results showed that PDGFD significantly increased the proliferation of bat embryonic fibroblasts, mouse preosteoblasts, and bat embryonic forelimb cells (*P* = 0.00022, *P* = 0.021, *P* = 0.00018; two-tailed Student's *t*-tests; Fig. 6i). Notably, both PDGFC and PDGFD more robustly enhanced the proliferation of bat embryonic forelimb cells compared to bat embryonic fibroblasts and mouse preosteoblasts, further suggesting that PDMPs, which secrete these two factors, may play crucial roles in the development of bat forelimbs.

Overall, these findings highlight the important roles of the PTN and PDGF signaling pathways in the development of bat forelimbs, suggesting that the interactions mediated by growth factors between PDMPs and bone-related cells are critical for the specialization of bat forelimbs during development.

## Discussion

The elongation of digits and expansion of interdigital membranes in forelimbs are essential for true flight in bats and represent unique morphological adaptations in mammals. In this study, we conducted single-cell transcriptomic analyses of developing bat limbs. Our findings confirmed that most MPs in developing bat limbs align with the established understanding that mammalian limbs initially form small

buds of mesenchymal cells[24]. Building on this, we clarified that bone-related cells originate from MPs with high *EBF2* expression in developing bat limbs. Histological comparisons with mouse forelimbs showed that bats have larger chondrocyte proliferation regions[3]. Our single-cell analyses further revealed more chondrocytes in developing bat forelimbs compared to bat hindlimbs and mouse forelimbs, highlighting distinct developmental trajectories in chondrogenesis and osteogenesis between developing bat forelimbs, bat hindlimbs, and mouse forelimbs. This suggests that prolonged chondrogenesis and delayed osteogenesis play a crucial role in the digit elongation of bat forelimbs.

To explore the gene regulatory mechanisms involved in growth factor production in PDMPs of developing bat forelimbs, we used SCENIC[25] to identify enriched transcription factors (TFs) by evaluating gene regulation network activity in each cell population. We identified 292 regulons with significantly enriched TF motifs across cell populations in developing bat forelimbs. Using K-means clustering, we categorized these regulons of each cell population into 10 clusters based on their activity levels (Supplementary Fig. 12a). This analysis revealed a cluster of 13 regulons with 13 master TFs, showing notably higher activity in PDMPs compared to other cell populations (Supplementary Fig. 12b). To address the sparse nature of single-cell RNA sequencing, we generated bulk RNA sequencing data of bat forelimbs and hindlimbs at CS18 and CS20 (Supplementary Data 5). We identified differentially expressed target genes (TGs) for each master TF between forelimbs and hindlimbs (Supplementary Data 6). Constructing the gene regulatory network for these TFs and TGs, we found a regulation between NR3C1 and PDGFD (Supplementary Fig. 12c), suggesting that NR3C1 potentially regulates the expression of the marker gene *PDGFD* of PDMPs. Functional enrichment analyses of these TFs and TGs revealed enriched Gene Ontology (GO) terms associated with membrane development, epithelium morphology, and cell proliferation (Supplementary Fig. 12d). These findings further support the role of PDMPs in promoting the proliferation and morphogenesis of other cell populations in developing bat forelimbs.

We also applied similar approaches to compare gene regulatory networks between developing bat forelimbs and hindlimbs and identified clusters that were up- and down-regulated in the developing bat forelimbs compared to hindlimbs (Supplementary Fig. 13). Each cluster contained different regulons with corresponding master TFs (Supplementary Fig. 14a). Using bulk RNA-seq data, we identified differentially expressed TGs for each master TF between forelimbs and hindlimbs and performed functional enrichment analyses (Supplementary Data 7). The enriched GO terms in the developing bat forelimbs related to cell proliferation (Supplementary Data 8) across various cell populations, including the Notch signaling pathway and positive regulation of cell cycle (Supplementary Fig. 14b). In contrast, the developing bat hindlimbs showed enrichment in terms linked to cell differentiation, such as the Wnt signaling pathway and ossification (Supplementary Fig. 14b). These findings further suggest that developing bat forelimbs have a greater capacity for chondrogenesis and cell proliferation, while the hindlimbs are more inclined towards chondrocyte maturation and ossification.

Notably, some of our findings align strongly with the results from bulk RNA-seq data of the developing bat wing[5]. For example, both studies revealed that some regulators, such as *MEIS2*, *HOXD10*, *HOXD11*, and *HOXD12*, crucial for digit specification and limb phenotypes, were highly expressed in developing bat forelimbs compared to hindlimbs. The two studies also showed that the canonical WNT/β-catenin signaling pathway appeared generally suppressed in developing bat forelimbs relative to hindlimbs. These findings underscore the importance of high expression of these genes and the canonical WNT/β-catenin pathway suppression in the developmental specialization of bat forelimbs. While bulk RNA-seq provides averaged gene expression across various cell types, single-cell RNA-seq offers insights into gene expression patterns within individual cell types. By analyzing our single-cell transcriptomic data, we clarified that *MEIS2* was highly expressed in MMPs and PDMPs, specific cell populations in developing bat forelimbs. Similarly, WNT/β-catenin signaling suppression was predominantly observed in CPs and osteoblasts in developing bat forelimbs compared to hindlimbs. Thus, single-cell RNA sequencing enables us to detail gene expression regulations contributing to bat wing formation at a more nuanced resolution.

It has long been proposed that bats retain interdigital membranes by suppressing apoptotic cell death, as developing phalanges in model organisms with free digits, such as mice and chicks, initially connect through mesodermal tissue that undergoes massive programmed cell death and recedes[26]. Experiments in mice indicate that BMP signaling regulates this apoptotic process, and its suppression prevents cell death[27]. However, maintaining interdigital membranes in bats requires more than suppressing BMP signaling; increased FGFs for cell survival are also necessary[4,28]. Our analyses of multiple-level data from developing bat limbs revealed that, alongside BMP-regulated cell death and FGF-controlled cell survival, cell proliferation mediated by PTN, PDGF, and Notch signaling pathways may also play important roles in the developmental specialization of bat forelimbs. Importantly, we identified a specific cell population, PDGFD+ MPs, in developing bat forelimbs compared to developing bat hindlimbs and mouse forelimbs. This cell population not only differentiates into interdigital membranes of bat forelimbs but also releases growth factors to enhance the proliferation of other cell populations in developing bat forelimbs. These findings suggest that a combination of molecular signaling pathways and new cell populations is vital for bat forelimb specialization during development.

In summary, our study offers a comprehensive census of single-cell transcriptomic profiles of developing bat forelimbs and hindlimbs. Despite challenges from using wild-caught, non-model organisms, we effectively addressed issues like sample size, biological variability, and tissue sampling. Through generating single-cell and bulk transcriptomic data, we identified the developmental origins and potential regulators involved in bat limb development. Our findings establish the cellular and molecular foundation for future studies to uncover the mechanisms behind the unique morphological innovations in bats.

## Methods

### Ethical regulations statement

All animal care and experimental protocols were approved by the Institutional Animal Care and Use Committee of the Kunming Institute of Zoology, Chinese Academy of Sciences (IACUC-PA-2021-06-013) and in accordance with the Animal Research: Reporting of In Vivo Experiments guidelines.

### Sample collection

Embryos of the Chinese horseshoe bat (*Rhinolophus sinicus*, RSI) and the Asian long-fingered bat (*Miniopterus fuliginosus*, MFU) were collected from a maternity roost in Kunming, Yunnan Province, China, annually from April to May. Mouse embryos of the C57BL/6 strain were provided by the Experimental Animal Center at the Kunming Institute of Zoology, Chinese Academy of Sciences. After euthanizing the animals by administering an overdose of isoflurane inhalation, their embryos were collected to determine the developmental stage based on phenotypic characteristics outlined in a previous study[9]. These embryos were cryopreserved in liquid nitrogen and stored at −80 °C for further experimentation.

### Skeletal staining and measurement

The embryos of bats and mice at various developmental stages were fixed overnight at 4 °C in 10% buffered formalin and then rinsed with flowing distilled water for over 48 h. Subsequently, these embryos were immersed in 1.5% hydrogen peroxide for 5–48 h until they

appeared translucent or white. The samples were then treated with 70% ethanol for 24 h, followed by anhydrous ethanol for an additional 24 h. They were stained for cartilage using 0.02% Alcian blue for 24–72 h and bones using 0.005% Alizarin red for 1–2 days. Finally, these samples were preserved in 100% glycerol for dissection and photography.

### Preparation and sequencing of single-cell RNA libraries

Following the SPLiT-seq methodology described by Rosenberg et al.[11], single-nucleus RNA-seq libraries were prepared from the forelimbs and hindlimbs of each RSI embryo at CS16, CS18, and CS20, as well as mouse embryo forelimbs at corresponding stages: E13.5, E14.5, and E16.5. Compared to other bat species, RSI has a greater population size in southwestern China, and consequently, their embryos are available more easily. Based on analyses of the developmental dynamics of bat wings, stages CS16, CS18, and CS20 are pivotal for digit elongation and interdigital membrane expansion in bat forelimbs. In brief, each sample was homogenized using Dounce homogenizers (Wheaton) with a loose pestle for 10 rounds, followed by 20 rounds with a tight pestle. The resulting homogenate was processed to obtain a uniform nuclear solution. Subsequently, the nuclear extract was filtered through a 40 μm cell strainer (Corning) into 1.5 ml tubes, which were then centrifuged to isolate the nuclei. The purified nuclei were then diluted to $1 \times 10^6$ nuclei/ml after quantification using a hemocytometer. The nuclei were evenly distributed among 48 wells of a 96-well plate, with each well containing 9000–12,000 nuclei. Subsequently, random hexamer and anchored poly(dT)15 barcoded reverse transcription primers were added to each of the 48 wells, respectively. The cells were then combined and split twice. Following this, the second- and third-round barcodes were attached. Finally, these barcoded cells were pooled together and then evenly splited into 16 libraries, which were subjected to sequencing on the Illumina platform to produce 150 bp pair-end reads. To validate the SPLiT-seq findings, single-nucleus RNA sequencing was performed on *R. sinicus* limbs at CS21 using the 10× Genomics platform, yielding 213 Gb clean reads (Supplementary Data 4). The isolation of nuclei and library preparation followed the guidelines outlined in the reference documents (CG000375 and CG000315) provided on the 10× Genomics website (https://www.10xgenomics.com/).

### Processing of raw scRNA-seq data

High-quality genomic data of RSI and its annotation file were obtained from the National Center for Biotechnology Information database (BioProject#: PRJNA1048078). This reference genome was constructed using 175.9 Gb of Nanopore long reads and 220.7 Gb of Hi-C reads. The resulting genome assembly covers ~2.04 Gb, with a scaffold N50 of 166.6 Mb and a BUSCO completeness score of ~94%. A comprehensive genome annotation approach, combining de novo and homology-based predictions and transcriptomic data from multiple tissues, identified 20,405 protein-coding genes. This high-quality genome assembly provides a reliable foundation for analyzing the single-cell RNA sequencing data of this species.

According to the SPLiT-seq methodology, a read was required to include three distinct barcode sequences. Any read lacking one of these barcode sequences was excluded from further analysis. After filtering, 288.4 Gb clean reads were obtained for the developing bat limbs, and 344.8 Gb clean reads for the developing mouse forelimbs (Supplementary Data 2). These clean reads were processed using the dropEst pipeline[29] to generate the gene-by-cell matrix. Simultaneously, the reads were aligned to either the *R. sinicus* reference genome (PRJNA1048078) or the mouse reference genome (mm10) using the STAR aligner[30] for annotations. Cells with fewer than 250 expressed genes, potential doublets with over 4000 expressed genes, and those containing more than 5% mitochondrial genes were excluded from further analyses. The dataset was analyzed using canonical correlation to mitigate potential batch effects arising from two different primer sets in SPLiT-seq,

followed by normalization using SCTransform in Seurat[31]. Following this, the dimension reduction was performed using UMAP (default parameters) based on the K-nearest-neighbor graph[32]. The marker genes for each cell population were determined using the FindAllMarkers function in Seurat. Cell populations were then annotated based on these marker genes and reference databases such as the Mouse Organogenesis Cell Atlas[13], ENRICHR[15], and METASCAPE[16]. Similarly, the snRNA-seq data from the 10× Genomics platform was processed using the approaches above. The clustering results from both platforms were integrated using the LabelTransfer function in Seurat.

### Analysis of developmental trajectory and cell-cell communications for scRNA-seq data

We performed single-cell trajectory analyses using monocle3[33] and scVelo[20]. The developmental trajectories of chondrogenesis and osteogenesis were aligned using CellAlign[22] to compare the expression dynamics in the relevant cell populations between the developing bat forelimbs and hindlimbs, as well as between the developing bat forelimbs and mouse forelimbs. Intercellular communication networks in the developing bat forelimbs, bat hindlimbs, and mouse forelimbs were inferred using CellChat[7] by examining the expression of signaling ligands and receptors in each cell population.

### Inference of single-cell regulatory networks

We used SCENIC[25] to infer the gene regulatory networks for scRNA-seq data. In brief, the read count matrix of scRNA-seq data was processed with GRNboost2[34] to identify TFs and their TGs by assessing the expression correlation of genes across cells, which are referred to as regulons. The regulons were further refined based on whether the TGs showed an enrichment of the binding motifs of the corresponding TF. The enrichment of individual cells was quantified using AUCell scores. To compare differentially activated TFs among cell populations, we evaluated and categorized the AUCell scores of all enriched regulons into 10 clusters using K-means clustering for each cell population. We then identified differentially activated clusters that satisfied two criteria based on the density distribution of AUC thresholds determined by SCENIC for each of the enriched regulons: (1) the mean AUCell score of the cluster >0.1, and (2) the difference in mean AUCell scores between clusters >0.5. The threshold indicates the cells in which the regulon is considered "active"[25]. Following this, we further refined the TGs that showed significant differences in expression patterns reflected by bulk RNA-seq datasets. Significantly differentially expressed genes in bulk RNA-seq datasets were determined using FPKM. Gene functional enrichment analysis was performed using Metascape[16].

### Whole-mount in situ hybridization

cDNA obtained from the adult bat kidney was used as bat-specific templates for WISH. Primers and probes were designed for genes *MEIS2*, *FREM1*, *SYNPO2*, *CSRNP3*, and *PDGFD* (Supplementary Data 9). The purified PCR products of *MEIS2*, *FREM1*, *SYNPO2*, *CSRNP3*, and *PDGFD* were inserted into the pGEM-3zf(+) vector between the HindIII and EcoRI sites, respectively. Plasmids were linearized using the corresponding restriction enzymes, and DIG-labeled probes were generated through in vitro transcription employing T7 or SP6 RNA polymerases. The WISH protocol followed previously established procedures[35]. Embryos were fixed in 4% PFA at 4 °C overnight and subjected to an ascending methanol series in PBT (25%, 50%, 75%, 2× 100% methanol for 15 min each step). After rehydration through a descending methanol series and two washes in PBT, the embryos were treated with proteinase K and then refixed in 4% PFA with 0.1% glutaraldehyde in PBT. Hybridization with DIG-labeled RNA probes was performed under controlled conditions (1.3× SSC pH 5, 50% formamide, 5 mM EDTA pH 8, 50 μg/mL torula RNA, 0.2% Tween-20, 0.5% CHAPS, and 100 μg/mL heparin) at 70 °C overnight. Subsequently, the embryos were rinsed with 1× TBST and 1× MSBT, followed by treatment

with a blocking reagent (Roche cat# 11096176001), and then incubated overnight with alkaline phosphatase-coupled anti-DIG antibody (Roche cat# 11093274910). After being washed with 1× MABT for 24 to 48 h, the embryos were stained with BM purple AP substrate (Roche cat# 11442074001) for 1–10 h. Finally, the embryos were treated with a gradient series of glycerol-PBT and dissected for photography using an optical microscope (Keyence, VHX-6000).

### Immunofluorescence of chondrocytes and osteoblasts

Paraffin sections of digit III of bat embryos at CS18 and CS20 were baked at 65 °C for 30 min, then cooled to room temperature. Deparaffinization was carried out using a xylene substitute for 15 min, repeated three times, followed by washes with an ethanol gradient (100%, 95%, 90%, 80%, and 70%) and water. Antigen retrieval was performed with 1× EDTA solution at 95 °C for 30 min, followed by six PBS washes. The sections were permeabilized using 0.1% Triton X-100 in PBS for 25 min and then washed with PBS. Blocking was done with 4% goat serum and 1% BSA for 1 h at room temperature. Primary antibodies, anti-COL2A1 (1:500, Santa Cruz, Cat. no. sc-52658) and anti-ABI3BP (1:200, Servicebio, Cat. no. GB114912), were respectively applied in the blocking solution and incubated overnight at 4 °C. After washing, EdU staining was performed with a kit (C10339, Invitrogen) for 40 min, followed by PBS washes. Secondary antibodies, Alexa Fluor™ Plus 647 (1:500, Invitrogen, Cat. no. A32728TR) and Cyanine3 (1:500, Invitrogen, Cat. no. A10520), were respectively incubated for 1 h at room temperature, and DAPI staining was done for 30 min. Finally, the slides were mounted with an anti-fade medium and imaged. Chondrocytes were marked using *COL2A1*, and osteoblasts were identified with *ABI3BP*.

### Stable HEK293T cell lines

To enhance the stability and repeatability of our experiments, we created HEK293T cell lines that stably express *PTN*, *PDGFC*, and *PDGFD*. In brief, the synthesized protein-coding sequences for *PTN*, *PDGFC*, and *PDGFD* were inserted into the pHBLV-CMV-MCS-3flag-EF1-ZsGreen-T2A-puromycin plasmid, respectively. Each of the three plasmids, along with packaging plasmids pSPAX2 and pMD2.G, were co-transfected into HEK293T cells using Lipofectamine 3000 (Thermo Fisher) at a ratio of 10:5:2. After 72 h, lentivirus particles were harvested and filtered using a 0.45 µm filter (Millipore). HEK293T cells were infected with the lentiviruses and selected with 2 µg/ml puromycin for 72 h to generate stable cell lines respectively expressing *PTN*, *PDGFC*, or *PDGFD*.

### Transwell co-culture assays

In a 6-well plate, cell-cell contact was obstructed by placing a 0.4 µm pore size polycarbonate membrane (Costar) in each well. RSI embryonic fibroblasts, MC3T3-E1 preosteoblasts, and cells from RSI embryonic forelimbs were respectively positioned below the membrane as target cells, while each of the donor cell lines (HEK293T-PTN/-PDGFC/-PDGFD) was placed on the membrane at 90% confluence. After co-culturing for 72 h, the target cells were evenly seeded per well in a 96-well plate. We added 20 µl of CellTiter 96 AQueous One Solution Reagent and 80 µl medium to each well and incubated the plate at 37 °C for 2 h in a humidified environment with 5% CO₂. The absorbance of the cells was measured at 490 nm using a 96-well plate reader. Cell proliferation capacity was assessed by comparing it against control groups transduced with empty vectors.

### Analyses of bulk RNA-seq data

The forelimbs and hindlimbs of RSI embryos at CS18 and CS20 were collected for bulk RNA sequencing on the Illumina platform, with each sample having two biological replicates. A total of 172 Gb bulk RNA-seq data were obtained (Supplementary Data 5). The raw reads from the bulk RNA-seq were trimmed using Trimmomatic[36] and mapped to the RSI reference genome with STAR to generate a table of read counts and FPKM values using RSEM[37]. Differentially expressed genes between bat forelimbs and hindlimbs were identified using a two-tailed Students' *t*-test with FPKM values.

### Statistics and reproducibility

No statistical method was used to predetermine sample size. No data were excluded from the analyses. The experiments were not randomized, and investigators were not blinded to allocation during experiments and outcome assessment. For morphological data, results were presented as mean ± SD with error bars, and individual data points were shown as single dots. Enrichment of cell population markers was assessed using two-sided Fisher's exact tests, and comparisons of cell proportions were performed using two-sided $\chi^2$ tests. Correlations of gene expression were calculated using Spearman correlation. For experimental data, results were also presented as mean ± SD with error bars and individual results as single dots; all experiments were independently repeated three times. The *P* values were from two-sided Student's *t*-tests, with a threshold of 0.05 for significance.

### Reporting summary

Further information on research design is available in the Nature Portfolio Reporting Summary linked to this article.

## Data availability

The scRNA-seq and RNA-seq data generated in this study have been deposited both in the NCBI database under accession code PRJNA1062446 and in the Genome Sequence Archive at the BIG data center, Chinese Academy of Sciences, under accession code PRJCA022686. Source data are provided with this paper.

## Code availability

All custom codes are available on Zenodo at https://doi.org/10.5281/zenodo.15628769[38].

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

## Acknowledgements

We thank the technical support from the Genome Center of Biodiversity of Kunming Institute of Zoology, Chinese Academy of Sciences. This work was funded by grants from the National Key R&D Program of China (2023YFA1800500), the National Natural Science Foundation of China (32192422, 32330014, U23A20452), the Yunnan Fundamental Research Projects (202201AS070058, 202102AA310055, 202301AT070285), and the CAS "Light of West China Program" (xbzg-zdsys-202113).

## Author contributions

Z.L. conceived and supervised the project. X.L. analyzed scRNA-seq data and constructed gene regulatory networks. J.B. obtained scRNA-seq data and morphological data of bat embryos. J.B., J.B.J., and C.J.S. performed in situ hybridization. J.B.J. conducted functional experiments. J.B.J., Q.L., P.C., and J.B. collected bat samples. Y.S.M. analyzed bulk RNA-seq data. Z.L. and X.L. wrote the paper with input from all authors.

## Competing interests

The authors declare no competing interests.
