## [Transparent Peer Review file · Nature Communications]

Single-cell expression profiling of bat wing development

Corresponding Author: Professor Zhen Liu

Version 0:

Reviewer comments:

Reviewer #1

(Remarks to the Author)

In the article 'Single-cell characterization of the developing bat wings' the authors characterize the developing bat wing using single-cell RNA-seq at different time points finding differences in cell populations between forelimb and hindlimb and also different gene expression and pathways. They also validate their finding using bat embryo in situs and cool transwell cell culture models. In general, the article could be a nice fit for the broad audience of Nature Communications but needs additional work. I have the following comments:

Major

-Single-cell results section needs more on whether RSI has a genome, how genes were annotated/called and how the comparison was then done for cell populations and to genes from what species. How many genes/transcripts were the authors not able to annotate? How many could be unique to bats? This could be an entire whole paragraph in results section.

-I appreciate the authors generating similar matching datasets for mouse. However, these were only done for forelimb and not hindlimb, making it in my mind not an exact comparison to some of the comparisons the authors are doing. I realize it is time consuming and costly but think it would be extremely helpful for the authors to generate similar datasets at similar time points for mouse hindlimb. Also, several of these mouse datasets should exist from various publications and the authors can use those instead or in addition. In general, it would be good for authors to compare to these other limb datasets to further confirm their results in another independent study/studies. It would also be good for the authors to compare results to the bulk RNA-seq that was done on the developing bat wing (Eckalbar Nature Genetics 2016) and showcase similarities but also advantages of their single-cell work over bulk. Would also mention this more in discussion.

Minor

-'Single-cell characterization of the developing bat wings' as there are many different single-cell ways to characterize (for example ATAC-seq) would change the title to clarify that this is RNA. Something like: 'Single-cell transcriptomic characterization of the developing bat wings'.

-Would mention bat species, number of cells sequenced and single-cell embryonic time points in abstract. Also, there are also interesting pathway findings in article that are not in the abstract. Would consider shrinking a bit the cell population text and adding a bit more on pathway to abstract.

-Would mention Xie et al. eLife (2020) in introduction that already showed delayed ossification in bat forelimb.

-'because the developmental coordination among local tissues is often shaped by interactions among cell progenitors, which drive the diversity and transitions in cell states during development7.' For better flow would move this sentence to the previous paragraph and also the single-cell addresses other things in previous paragraph also, so would have this paragraph begin with this sentence: 'To address these issues, we conducted an in-depth single-cell analysis of developing bat limbs. Our systematic investigation...'

-Would change last sentence of intro 'We suggested that...' to 'Our data suggests that...'

-‘RSI forelimbs and hindlimbs at CS16, CS18, and CS20’. Would provide at least 1-2 sentences saying why this bat species was used (and not MF) and these time points for the single cell work.

-‘Next, we compared the abundance of each cell population between the developing bat forelimbs and hindlimbs’ assuming this is for all populations. If so would make it clear. Would then separate the next section ‘To unravel the developmental dynamics’ looking at different time points to be an independent paragraph so easier for reader to follow.

-‘The proportions of these two cell populations were significantly higher in bat forelimbs (7.2% and 11.5%) compared to hindlimbs (0.9% and 0.7%) ($P < 0.0001$, $P < 0.0001$, χ^2 tests; Fig. 4c).’ Were these normalized in some way to total number of cells from each tissue/time point. More text on if this was done and how is needed in results section.

-‘To validate the suggestion above, we systematically compared the temporal gene’ Would mention in a few words what ‘the suggestion above’ is to make it easier for the reader to understand.

-‘HEK293T cells with stable PTN overexpression’ more text in results on how this cell line was made and stably expresses PTN and same later for PDGFC.

-Figure 1 only shows skeletal preps of *Rhinolophus sinicus* but later shows both *Rhinolophus sinicus* and *Miniopterus fuliginosus* in the graphs. To make it easier for the reader, would also add skeletal preps of and *Miniopterus fuliginosus* to figure.

Reviewer #2

(Remarks to the Author)

Lui et al. use scRNA seq analysis to study the developmental stages critical for bat forelimb digit elongation and flight membrane development. This analysis identifies a forelimb specific mesenchymal cell population that is characterized by high PDGFD ligand expression and provides evidence that this PDMP population differentiates into the interdigital flight membranes. In addition, the authors provide evidence that this cell population releases signals that promote proliferation of osteogenic cells contributing to digit elongation. In contrast to bat hindlimbs (and mouse limbs), the developmental time period of chondrogenesis is prolonged while osteogenesis is delayed during digit elongation and flight membrane formation in bat forelimb buds. This is a potentially interesting study but there are several major issues that need to be addressed by additional experiments.

1. Many of the conclusions are based on the timeseries of the scRNA seq analysis. While this is not per se a shortfall, the cell populations need to be characterized in much more depth to e.g. identify the genes whose expression is enriched in one population versus the others in an unbiased and quantitative manner. Also there is a need for functionally relevant experimental follow-up analysis of the key cell populations. Also the conclusions concerning cell proliferation and delayed osteogenesis need to be substantiated by experimental analysis of cell proliferation and osteogenesis using the appropriate RNA probes, antibodies or labeling proliferating cell in situ at different stages of bat forelimb development.

2. Linked to comment (1), only one marker gene each, namely *Meis2* and *PDGFD* for the MMP and PDMP populations, respectively, are analyzed by RNA in situ hybridization at one stage. To be informative, these and additional marker genes for each population should be analyzed at all relevant stages to get insight into the population dynamics and reveal or exclude potential heterogeneity and/or regionalization of these two populations and proliferating cells. This is of particular importance with respect to potential antero-posterior and proximo-distal patterning that is observed during normal mouse limb development. In short, such additional analysis will provide insight into the possible spatial regionalization and /or heterogeneity of the newly identified MMP and PDMP populations. Is there differences in their antero-posterior and/or proximo-distal pattern as observed for digits or they are homogenous?

3. The transwell experiments must be done using bat forelimb cells, ideally PDMP cells rather than bat embryonic fibroblasts to reach firm and functionally relevant conclusions.

4. It is not clear what is the purpose of the signaling pathway analysis shown in Fig. 6a. It seems a mere listing of upregulated (= upactivated?) pathways.

5. The same applies to most of the analysis shown in Fig. 7. What are the conclusions from this analysis beyond identifying sets of genes/GRNs and pathways /open chromatin. Each of these analysis requires insightful follow-ups. The selection of *TBX15* binding sites over others is not clear.

In summary, the rationale for including different types of bioinformatics analyses in the manuscript is not clear and often appears as mere listings. The relevance of the observed differences and enrichments is not put in a functional context.

In its current state the study does not provide much insights beyond the identification of the bat forelimb specific PDMP population, which is important but must be complemented by vigorous experimental analysis of this and the MMP population. Finally, the manuscript requires significant restructuring to allow the reader to follow the logic of analysis.

Reviewer comments:

Reviewer #1

(Remarks to the Author)

The authors have very nicely responded to all my comments. I have no further comments.

Reviewer #2

(Remarks to the Author)

Lyu and coworkers have performed the necessary revisions to address both reviewers comments. This has very significantly improved the manuscript and it will likely be a nice study for Nature Communication. Having stated this, a number of issues remain that require attention before final consideration of the manuscript for Nature Communications is possible.

Major point

This reviewer still has major concerns and reservations with respect to the bioinformatics analyses and its implications for bat forewing development shown in Figure 7 (and Supplementary Figs. 11-15). This represents an in depth analysis of the different RNA-seq datasets, but it remains weak with respect to drawing firm conclusions. In particular, there is no experimental follow-up that provides experimental insights and evidence supporting the relevance of these networks with respect to bat forewing development. For example, it is not surprising that Hox transcriptional regulators are enriched and control proliferation (the latter is long established). Furthermore, the gene network analysis pinpoints the Notch pathway and Tbx15 as potentially relevant to bat wing development, but there is no spatio-temporal expression or experimental analysis providing supporting evidence for specific functions during bat forewing development. At least one previous study has shown that there is a second phase of the Shh-Grem1/Bmp-Fgf8 signaling system active in the interdigit tissue that underlies prolonged bat forewing outgrowth and digit development (Hockman et al. 2008, <https://doi.org/10.1073/pnas.0805308105>). Unexpectedly, this system is not detected by the current analysis. Suppression of the Wnt pathway in bat forewings is rather counterintuitive in light of Wnt being a key component of the Turing type system that controls digit patterning and development in mice (Raspopovic et al. 2014 DOI: 10.1126/science.1252960).

There are two options in this reviewer's opinion: 1. remove this analysis as there is no experimental follow-up such as spatio-temporal analysis of the differential expression of Notch pathway genes, Tbx15 and no firm functional conclusions to be reached at this stage. 2. Alternatively, (part of) this bioinformatics analysis could be described prior to focusing on the two cell-populations of interest (MMPs and DDMPs), the increase in chondrogenesis and delay in osteogenesis. The rationale for this is that these latter findings are the most interesting and conclusive parts of the study. In parallel to modifying the result section the discussion needs to be restructured.

Minor points (following text and figures)

Title- another possibility would be: "Single cell expression profiling of bat wing development"

Abstract: line 28 - define: ...Carnegie stages (CS) 16, 18, 20

Typos: line 26: ...remains...; line 34: ...in combination with ATAC...

Fig. 1b: please label the panels "Digit 3 forelimb" and "Digit 3 hindlimb" for clarity

Fig. 4e and Supplementary Fig. 5: label the digits I-V in the panels of both fore- and hindlimbs.

Line 213-218: the two sentences state pretty much the same- this reviewer understands two methods were used (Monocle 3 and scVelo) arriving at the same conclusion. This can be stated in one sentence.

Line 213-229: the description of Supplementary Figs 6 and 7 is in parts confusing and would gain from improvement and better structuring.

Line 243-247: to visually validate the conclusion "that calcified bones appeared in hindlimb phalanges, whereas no such calcifications were detected in forelimb phalanges" an additional Supplementary Fig. showing representative high magnifications needs to be included.

Line 254(error): Fig. 5g not 5f

Line 258/259 (error): Fig. 5f not 5g

Fig. 6e: transwell experiments are tricky and with n=3 biological replicates it is difficult to reach statistically solid conclusions for PTN, but this reviewer agrees that these experiments show a trend to increased proliferation. This should be considered when describing and discussing these results. Interestingly, the response observed following PDGFC and PDGFD treatments of embryonic forelimb cells appear as very robust - this is most relevant and needs to be pointed out better in the description.

Version 2:

Reviewer comments:

Reviewer #2

(Remarks to the Author)

The authors have now addressed all remaining comments of this reviewer which has improved the clarity and logic structure of the study. The manuscript can now be accepted in this reviewer's opinion.

Response to the reviewers

We are grateful to the reviewers for their constructive comments, which have helped improve our manuscript. Below please find our point-to-point response (in blue).

Reviewer #1

Comment 1

In the article ‘Single-cell characterization of the developing bat wings’ the authors characterize the developing bat wing using single-cell RNA-seq at different time points finding differences in cell populations between forelimb and hindlimb and also different gene expression and pathways. They also validate their finding using bat embryo in situs and cool transwell cell culture models. In general, the article could be a nice fit for the broad audience of Nature Communications but needs additional work.

Response

We thank the reviewer for the positive evaluation.

I have the following comments:

Comment 2

Single-cell results section needs more on whether RSI has a genome, how genes were annotated/called and how the comparison was then done for cell populations and to genes from what species. How many genes/transcripts were the authors not able to annotate? How many could be unique to bats? This could be an entire whole paragraph in results section.

Response

We obtained the genomic data of the Chinese horseshoe bat (*Rhinolophus sinicus*, RSI) and its associated annotation file from the National Center for Biotechnology Information database (BioProject#: PRJNA1048078). The genomic annotation file and accompanying documentation indicate that this reference-level genome was constructed using 175.9 Gb of Nanopore long reads and 220.7 Gb of Hi-C reads. The resulting genome assembly covers ~2.04 Gb, achieving a scaffold N50 of 166.6 Mb and a BUSCO completeness score of ~94%. For genome annotation, a comprehensive approach, including de novo and homology-based predictions, as well as incorporating transcriptomic data from multiple tissues, was employed and identified a total of 20,405 protein-coding genes. This high-quality de novo genome assembly of the Chinese horseshoe bat serves as a reliable foundation for analyzing our single-cell RNA sequencing data of RSI. We have included a paragraph detailing the source and quality of the RSI genomic data for added clarity (page 19, paragraph 2).

Comment 3

I appreciate the authors generating similar matching datasets for mouse. However,

these were only done for forelimb and not hindlimb, making it in my mind not an exact comparison to some of the comparisons the authors are doing. I realize it is time consuming and costly but think it would be extremely helpful for the authors to generate similar datasets at similar time points for mouse hindlimb. Also, several of these mouse datasets should exist from various publications and the authors can use those instead or in addition. In general, it would be good for authors to compare to these other limb datasets to further confirm their results in another independent study/studies. It would also be good for the authors to compare results to the bulk RNA-seq that was done on the developing bat wing (Eckalbar Nature Genetics 2016) and showcase similarities but also advantages of their single-cell work over bulk. Would also mention this more in discussion.

Response

This is an excellent suggestion. Following the suggestion, we retrieved single-cell transcriptomic data of developing mouse hindlimbs from two previous studies (PMID: 38057666 and PMID: 31874220). Their developmental stages corresponded to those of developing bat limbs and mouse forelimbs in our study. Upon comparison, we found consistent evidence with the developing mouse forelimbs: no cell population in the developing mouse hindlimbs was closely associated with MMPs and PDMPs. This finding further supports the robustness of our results, indicating that MMPs and PDMPs were specific cell populations in the developing bat forelimbs. We have included these findings in the revised manuscript (**Supplementary Fig. 3**; page 7, paragraph 2).

Following the suggestion, we compared our findings with those from bulk RNA-seq data of the developing bat wing (Eckalbar et al. Nature Genetics 2016). We found a strong consistency between our results and theirs. For example, regulators significantly involved in digit specification and limb phenotypes, such as MEIS2, HOXD10, HOXD11, and HOXD12, were highly expressed in developing bat forelimbs compared to developing bat hindlimbs. Additionally, the canonical WNT signaling pathway appeared to be generally suppressed in the developing bat forelimbs relative to developing bat hindlimbs. These consistent findings highlight the critical roles of the high expression levels of these *HOXD* genes and the suppression of the canonical WNT signaling pathway in the developmental specialization of bat forelimbs. While bulk RNA sequencing provides averaged gene expression across various cell types, single-cell RNA sequencing offers insights into gene expression patterns within individual cell types. By integrating our single-cell RNA sequencing data, we have clarified that MEIS2 is highly expressed in MMPs and PDMPs, specific cell populations in developing bat forelimbs. Similarly, HOXD10 shows high activity in chondrocyte precursors (CPs), and the suppression of WNT signaling is predominantly observed in CPs and osteoblasts of developing bat forelimbs, when compared to the developing bat hindlimbs. Consequently, single-cell RNA sequencing enables us to detail gene expression regulations contributing to bat wing formation at a more nuanced resolution. We have incorporated a discussion that examines the comparison of our results with the previous study, as well as the advantages our single-cell RNA sequencing work holds over bulk RNA sequencing

(page 15; paragraph 2).

Comment 4

‘Single-cell characterization of the developing bat wings’ as there are many different single-cell ways to characterize (for example ATAC-seq) would change the title to clarify that this is RNA. Something like: ‘Single-cell transcriptomic characterization of the developing bat wings’.

Response

We have revised the title as the reviewer suggested.

Comment 5

Would mention bat species, number of cells sequenced and single-cell embryonic time points in abstract. Also, there are also interesting pathway findings in article that are not in the abstract. Would consider shrinking a bit the cell population text and adding a bit more on pathway to abstract.

Response

In the revised abstract, we have included the bat species, the number of cells sequenced, single-cell embryonic time points, and key pathway findings. Additionally, we have condensed the text related to cell populations.

Comment 6

Would mention Xie et al. eLife (2020) in introduction that already showed delayed ossification in bat forelimb.

Response

We have cited Xie et al. eLife (2020) and mentioned their results regarding the delayed ossification in the developing bat forelimbs.

Comment 7

‘because the developmental coordination among local tissues is often shaped by interactions among cell progenitors, which drive the diversity and transitions in cell states during development.’ For better flow would move this sentence to the previous paragraph and also the single-cell addresses other things in previous paragraph also, so would have this paragraph begin with this sentence: ‘To address these issues, we conducted an in-depth single-cell analysis of developing bat limbs. Our systematic investigation...’

Response

Following the suggestion, we have relocated the sentence to the previous paragraph and began this paragraph with the sentence recommended by the reviewer.

Comment 8

Would change last sentence of intro ‘We suggested that...’ to ‘Our data suggests that...’

Response

We have modified this sentence as recommended.

Comment 9

‘RSI forelimbs and hindlimbs at CS16, CS18, and CS20’. Would provide at least 1-2

sentences saying why this bat species was used (and not MF) and these time points for the single cell work.

Response

Compared to other bat species, such as the eastern bent-winged bat (*Miniopterus fuliginosus*, MFU), the Chinese horseshoe bat (*Rhinolophus sinicus*, RSI) has a larger population size in the southwest region of China, where our institute is located. This abundance makes their embryos more readily available for study. Based on analyses of the developmental dynamics of bat wings, CS16, CS18, and CS20 were identified as pivotal developmental stages for digit elongation and interdigital membrane expansion in bat forelimbs. We have included the information in the Methods section (page 19, paragraph 1).

Comment 10

‘Next, we compared the abundance of each cell population between the developing bat forelimbs and hindlimbs’ assuming this is for all populations. If so would make it clear. Would then separate the next section ‘To unravel the developmental dynamics’ looking at different time points to be an independent paragraph so easier for reader to follow.

Response

It is true for all cell populations. We have revised this sentence as follows for clarity and created a new paragraph for the subsequent section.

“Next, we compared the abundance of each of the 16 cell populations between the developing bat forelimbs and hindlimbs across different developmental stages”

Comment 11

‘The proportions of these two cell populations were significantly higher in bat forelimbs (7.2% and 11.5%) compared to hindlimbs (0.9% and 0.7%) ($P < 0.0001$, $P < 0.0001$, χ^2 tests; Fig. 4c).’ Were these normalized in some way to total number of cells from each tissue/time point. More text on if this was done and how is needed in results section.

Response

Different tissues contain varying cell types and numbers available for single-cell RNA sequencing. To ensure a fair comparison of the abundance of each cell population between the developing bat forelimbs and hindlimbs, we normalized the cell number for each cell population to the total number of cells from each tissue across different developmental stages, as recommended by the reviewer. We have included this explanation in the results section (page 6, paragraph 2).

Comment 12

‘To validate the suggestion above, we systematically compared the temporal gene’ Would mention in a few words what ‘the suggestion above’ is to make it easier for the reader to understand.

Response

“the suggestion above” means “increased chondrogenesis and delayed osteogenesis in the developing bat forelimbs”. We have modified the sentence as follows to clarify

the phrase:

“To validate the increased chondrogenesis and delayed osteogenesis in the developing bat forelimbs compared to bat hindlimbs, we systematically compared...”

Comment 13

‘HEK293T cells with stable PTN overexpression’ more text in results on how this cell line was made and stably expresses PTN and same later for PDGFC.

Response

After synthesizing target genes *PTN*, *PDGFC*, and *PDGFD*, each was inserted into the backbone plasmid pHBLV-CMV-MCS-3flag-EF1-ZsGreen-T2A-Puromycin. To produce lentivirus particles, these constructs, along with the packaging plasmids pSPAX2 and pMD2.G, were co-transfected into HEK 293T cells using Lipofectamine 3000 at a ratio of 10:5:2. After 72 hours of transfection, the lentivirus particles were harvested and filtered through a 0.45 µm filter. HEK 293T cells were then infected with the filtered lentiviruses and treated with 2 µg/ml puromycin for 72 hours to establish cell lines stably expressing *PTN*, *PDGFC*, or *PDGFD*. We have added further details to the Methods section and included the relevant information in the Results section (page 22, paragraph 1).

Comment 14

Figure 1 only shows skeletal preps of *Rhinolophus sinicus* but later shows both *Rhinolophus sinicus* and *Miniopterus fuliginosus* in the graphs. To make it easier for the reader, would also add skeletal preps of and *Miniopterus fuliginosus* to figure.

Response

Following the suggestion, we have included skeletal preps of *Miniopterus fuliginosus*, as well as mouse, in Figure 1.

Reviewer #2

Comment 1

Lui et al. use scRNA seq analysis to study the developmental stages critical for bat forelimb digit elongation and flight membrane development. This analysis identifies a forelimb specific mesenchymal cell population that is characterized by high PDGFD ligand expression and provides evidence that this PDMP population differentiates into the interdigital flight membranes. In addition, the authors provide evidence that this cell population releases signals that promote proliferation of osteogenic cells contributing to digit elongation. In contrast to bat hindlimbs (and mouse limbs), the developmental time period of chondrogenesis is prolonged while osteogenesis is delayed during digit elongation and flight membrane formation in bat forelimb buds. This is a potentially interesting study but there are several major issues that need to be addressed by additional experiments.

Response

We are pleased that the reviewer finds the topic of our study interesting. Following suggestions, we have conducted additional experiments to address the relevant issues. The results from these experiments further reinforce our conclusions. Please find the detailed information below.

Comment 2

Many of the conclusions are based on the timeseries of the scRNA seq analysis. While this is not per se a shortfall, the cell populations need to be characterized in much more depth to e.g. identify the genes whose expression is enriched in one population versus the others in an unbiased and quantitative manner. Also there is a need for functionally relevant experimental follow-up analysis of the key cell populations. Also the conclusions concerning cell proliferation and delayed osteogenesis need to be substantiated by experimental analysis of cell proliferation and osteogenesis using the appropriate RNA probes, antibodies or labeling proliferating cell in situ at different stages of bat forelimb development.

Response

This is an excellent suggestion. To gain a deeper understanding of the cell populations in developing bat limbs, we followed the suggestion to identify genes that are highly expressed in each cell population and conducted functional enrichment analyses using the Jensen TISSUES and Metascape databases. The results showed that the enriched tissue types and functional categories aligned with our cell population annotations. This alignment reinforces the reliability of our annotations and provides greater insight into the roles these cell populations play in the specialization of bat forelimbs. These findings have been included in the revised manuscript (**Figs. 2d** and **2e**; page 6, paragraph 1).

We have conducted additional experimental follow-up analyses on the key cell populations. For example, we selected several additional marker genes for the MMPs and PDMPs based on our scRNA-sequencing, and performed RNA in situ hybridization analyses across different developmental stages. These analyses

verified the specificity of these cell populations in the developing bat forelimbs. Additionally, using transwell assays, we confirmed that the regulators PDGFD and PDGFC, released by PDMPs, can enhance cell proliferation in bat embryonic forelimbs. These experiments further support the robustness of our conclusions. Detailed information is provided in the response to the next comments.

To confirm the differing proliferation rates of chondrocytes between developing bat forelimbs and hindlimbs, we examined embryonic samples at CS18 and CS20, treated with EdU via intraperitoneal injection to label cell proliferation. Using COL2A1 as a chondrocyte marker, we performed immunohistochemistry on the most elongated digit III (**Fig. 5e**). The results showed significantly higher EdU intensities in bat forelimb chondrocytes compared to hindlimbs at both stages ($P = 0.0024$, $P = 0.015$; two-tailed Student's *t*-tests; **Fig. 5f**). Additionally, to investigate differences in osteogenesis between developing bat forelimbs and hindlimbs, we selected ABI3BP as an osteoblast marker based on our scRNA-sequencing. The results indicated a significant increase in ABI3BP-positive cells in developing bat hindlimbs at CS20 compared to forelimbs ($P = 0.04$, two-tailed Student's *t*-test; **Fig. 5g and 5h**). These findings provide strong evidence of enhanced cell proliferation and delayed osteogenesis in developing bat forelimbs compared to hindlimbs. We have included these results in the revised manuscript (page 10, paragraph 1).

Comment 3

Linked to comment (1), only one marker gene each, namely *Meis2* and *PDGFD* for the MMP and PDMP populations, respectively, are analyzed by RNA in situ hybridization at one stage. To be informative, these and additional marker genes for each population should be analyzed at all relevant stages to get insight into the population dynamics and reveal or exclude potential heterogeneity and/or regionalization of these two populations and proliferating cells. This is of particular importance with respect to potential antero-posterior and proximo-distal patterning that is observed during normal mouse limb development. In short, such additional analysis will provide insight into the possible spatial regionalization and/or heterogeneity of the newly identified MMP and PDMP populations. Is there differences in their antero-posterior and/or proximo-distal pattern as observed for digits or they are homogenous?

Response

Following the suggestion, we conducted RNA in situ hybridization analyses of the marker gene *MEIS2* for MMPs and *PDGFD* for PDMPs at developmental stages CS16, CS18, and CS20. The results across these relevant stages consistently indicated that MMPs and PDMPs were specific cell populations within developing bat flight membranes. Additionally, both marker genes showed high expression in the interdigital membranes between digits III, IV, and V, which were the most stretched digits compared to hindlimbs, rather than between digits I, II, and III. An antero-posterior pattern was suggested for the developing interdigital membranes. To confirm these findings, we also employed *FREMI* and *SYNPO2*, which were highly expressed in MMPs, and *CSRNP3*, which was highly expressed in PDMPs, as

additional markers in our RNA in situ hybridization analyses. These analyses across different marker genes yielded highly consistent results, which have been included in the revised manuscript (**Fig. 4e** and **Supplementary Fig. 5**; page 8, paragraph 3).

Comment 4

The transwell experiments must be done using bat forelimb cells, ideally PDMP cells rather than bat embryonic fibroblasts to reach firm and functionally relevant conclusions.

Response

We completely agree with the reviewer's suggestion to use PDMPs for transwell experiments to ensure firm and functionally relevant conclusions. However, the number of PDMPs is very limited, with ~6000 cells in a bat embryonic forelimb at CS18. More importantly, commercial antibodies specific to bats are unavailable. We tested PDMP marker antibodies, including anti-PDGFD and anti-ITGA4, which are designed for mice and humans, but none were effective for isolating PDMPs using flow cytometry. Therefore, we used cells from bat embryonic forelimbs to perform the transwell assay, as suggested. The results indicated a significant increase in cell proliferation of bat embryonic forelimbs compared to control groups under the overexpression of PTN, PDGFD, and PDGFC ($P = 0.01$, $P = 0.00018$, $P = 1.65E-06$; two-tailed Student's *t*-tests). The findings further support the robustness of our conclusions and have been included in the revised manuscript (**Figs. 6g** and **6h**).

Comment 5

It is not clear what is the purpose of the signaling pathway analysis shown in Fig. 6a. It seems a mere listing of upregulated (= upactivated?) pathways.

Response

This figure conveniently allows readers to view the up-activated signaling pathways in various cell populations within the developing bat forelimbs compared to hindlimbs, where "up-activated" refers to the upregulation of genes involved in these pathways. We also could clearly show that the PTN signaling pathway was the most up-activated among the pathways in the chondrocyte progenitors of bat embryonic forelimbs. We have modified the relevant sentence to clarify the purpose of this figure (page 11, paragraph 2).

Comment 6

The same applies to most of the analysis shown in Fig. 7. What are the conclusions from this analysis beyond identifying sets of genes/GRNs and pathways/open chromatin. Each of these analyses requires insightful follow-ups. The selection of TBX15 binding sites over others is not clear.

Response

The purpose of Fig. 7 is to illustrate the gene regulations underlying the developmental specialization of bat forelimbs. We have clarified Fig. 7 through several modifications. First, we annotated the gene regulation network up-activated in PDMPs, highlighting its close relationship with membrane development, epithelium morphology, and cell proliferation. Second, we included the gene regulation network of the Notch signaling pathway, demonstrating that TBX15 is a

hub TF with the most connections and potentially regulates several *HOXD* members, such as *HOXD3*, *HOXD8*, *HOXD10*, and *HOXD11*. Indeed, our ATAC-seq data revealed three TBX15 binding motifs within the *HOXD* gene cluster, which were more accessible in developing bat forelimbs than hindlimbs. Third, due to a TBX15 binding site located within the region of *HOXD3*, we overexpressed *HOXD3* in HEK293T cells and found that this overexpression significantly enhanced cell proliferation after 48 and 72 hours of culture ($P = 0.0021$, $P = 6.07E-06$; two-tailed Student's *t*-tests). We have included these modifications in the revised **Fig. 7**, and moved **Fig. 7a** and **Fig. 7b** to the supplementary files.

Comment 7

In summary, the rationale for including different types of bioinformatics analyses in the manuscript is not clear and often appears as mere listings. The relevance of the observed differences and enrichments is not put in a functional context. In its current state the study does not provide much insights beyond the identification of the bat forelimb specific PDMP population, which is important but must be complemented by vigorous experimental analysis of this and the MMP population. Finally, the manuscript requires significant restructuring to allow the reader to follow the logic of analysis.

Response

We appreciate the reviewer's constructive comments, which have greatly improved the quality of our manuscript. In response to the suggestions, we conducted additional experiments that further support the robustness of our conclusions. By using two additional markers, we have verified PDMPs and MMPs are specific to the developing bat forelimbs. Furthermore, these progenitors may not only differentiate into bat membranes but also release growth factors that enhance the proliferation of other cell types, highlighting the important roles of cell-cell crosstalk in the developing bat forelimbs. Additionally, we identified TBX15 as a key transcription factor involved in Notch signaling, which upregulates the expression of *HOXD* genes crucial for the elongation of bat forelimbs. We functionally verified that the overexpression of *HOXD3* increased cell proliferation. Please refer to the detailed responses above. These results, combined with our previous findings, provide valuable insights into the cellular and molecular mechanisms underlying the developmental specialization of bat forelimbs. We have restructured the manuscript to enhance the logic flow of our analyses for easier comprehension by readers.

Responses to the reviewers

Reviewer #1

Comment

The authors have very nicely responded to all my comments. I have no further comments.

Response

We thank the reviewer for reviewing our revised manuscript.

Reviewer #2

Comment 1

Lyu and coworkers have performed the necessary revisions to address both reviewers comments. This has very significantly improved the manuscript and it will likely be a nice study for Nature Communication. Having stated this, a number of issues remain that require attention before final consideration of the manuscript for Nature Communications is possible.

Response

We appreciate the reviewer for carefully evaluating our revised manuscript and providing constructive comments, which have greatly contributed to improving the quality of our work.

Comment 2

This reviewer still has major concerns and reservations with respect to the bioinformatics analyses and its implications for bat forewing development shown in Figure 7 (and Supplementary Figs. 11-15). This represents an in depth analysis of the different RNA-seq datasets, but it remains weak with respect to drawing firm conclusions. In particular, there is no experimental follow-up that provides experimental insights and evidence supporting the relevance of these networks with respect to bat forewing development. For example, it is not surprising that Hox transcriptional regulators are enriched and control proliferation (the latter is long established). Furthermore, the gene network analysis pinpoints the Notch pathway and Tbx15 as potentially relevant to bat wing development, but there is no spatio-temporal expression or experimental analysis providing supporting evidence for specific functions during bat forewing development. At least one previous study has shown that there is a second phase of the Shh-Grem1/Bmp-Fgf8 signaling system active in the interdigit tissue that underlies prolonged bat forewing outgrowth and digit development (Hockman et al. 2008, <https://doi.org/10.1073/pnas.0805308105>). Unexpectedly, this system is not detected by the current analysis. Suppression of the Wnt pathway in bat forewings is rather counterintuitive in light of Wnt being a key component of the Turing type system that controls digit patterning and development in mice (Raspopovic et al. 2014 DOI: 10.1126/science.1252960). There is two options in this reviewers opinion: 1. remove this analysis as there is no experimental follow-up such as spatio-temporal analysis of the differential expression

of Notch pathway genes, Tbx15 and no firm functional conclusions to be reached at this stage. 2. Alternatively, (part of) this bioinformatics analysis could be described prior to focusing on the two cell-populations of interest (MMPs and DDMPs), the increase in chondrogenesis and delay in osteogenesis. The rationale for this is that these latter findings are the most interesting and conclusive parts of the study. In parallel to modifying the result section the discussion needs to be restructured.

Response

Following the suggestion, we have removed Figure 7c-g and its corresponding supplementary figures 13 and 15 from the revised manuscript. As requested by Reviewer 1, we compared our findings from the bulk RNA-seq data analyses with those of a previous study (Eckalbar et al. 2016; DOI: 10.1038/ng.3537). Therefore, we have relocated the original Figure 7a and 7b to the supplementary materials (**Supplementary Fig. 14**).

The canonical WNT/ β -catenin signaling pathway was found to be suppressed in developing bat forelimbs compared to hindlimbs. In other words, this pathway was more highly activated in developing bat hindlimbs than in forelimbs. This result is highly consistent not only with the previous study by Eckalbar et al. (2016; DOI: 10.1038/ng.3537), but also with the Science paper by Raspopovic et al. (2014 DOI: 10.1126/science.1252960), as referenced by the reviewer. Raspopovic et al. showed that WNT ligands can repress chondrogenesis, and that inhibition of β -catenin in the limb results in expansion of proliferative regions. To improve clarity, we have replaced the general WNT pathway with the canonical WNT/ β -catenin pathway throughout the revised manuscript.

Comment 2

Title- another possibility would be: “Single cell expression profiling of bat wing development”

Response

We have revised the title as suggested.

Comment 3

Abstract: line 28 - define: ...Carnegie stages (CS) 16, 18 ,20

Response

Corrected.

Comment 4

Typos: line 26: ...remains...; line 34: ...in combination with ATAC...

Response

Corrected.

Comment 5

Fig. 1b: please label the panels “Digit 3 forelimb” and “Digit 3 hindlimb” for clarity

Response

We have added labels to Fig. 1b to clearly indicate “Digit III of forelimb” and “Digit III of hindlimb”.

Comment 6

Fig.4e and Supplementary Fig. 5: label the digits I-V in the panels of both fore- and hindlimbs.

Response

We have labeled the digits in the figures as suggested.

Comment 7

Line 213-218: the two sentences state pretty much the same- this reviewer understands two methods were used (Monocle 3 and scVelo) arriving at the same conclusion. This can be stated in one sentence.

Response

We have revised the sentences as suggested.

Comment 8

Line 213-229: the description of Supplementary Figs 6 and 7 is in parts confusing and would gain from improvement and better structuring.

Response

We have revised these sentences for clarity in the revised manuscript.

Comment 9

Line 243-247: to visually validate the conclusion “that calcified bones appeared in hindlimb phalanges, whereas no such calcifications were detected in forelimb phalanges” an additional Supplementary Fig. showing representative high magnifications needs to be included.

Response

We have included an additional supplementary figure with representative high magnifications in the revised manuscript to visually validate this conclusion (**Supplementary Fig. 9**).

Comment 10

Line 254(error): Fig. 5g not 5f

Response

Corrected.

Comment 11

Line 258/259 (error): Fig. 5f not 5g

Response

Corrected.

Comment 12

Fig. 6e: transwell experiments are tricky and with n=3 biological replicates it is difficult to reach statistically solid conclusions for PTN, but this reviewer agrees that these experiments show a trend to increased proliferation. This should be considered when describing and discussing these results. Interestingly, the response observed following PDGFC and PDGFD treatments of embryonic forelimb cells appear as very robust - this is most relevant and needs to be pointed out better in the description.

Response

We have replaced “significant” with “a general increasing trend” in the description of the PTN treatment result. In addition, we have added the following sentence to the description of the results for PDGFC and PDGFD treatments:

“Notably, both PDGFC and PDGFD more robustly enhanced the proliferation of bat embryonic forelimb cells compared to bat embryonic fibroblasts and mouse preosteoblasts, further suggesting that PDMPs, which secrete these two factors, may play crucial roles in the development of bat forelimbs.” (page xxx, paragraph xxx)

Responses to the reviewers

Reviewer #2

Comment

The authors have now addressed all remaining comments of this reviewer which has improved the clarity and logic structure of the study. The manuscript can now be accepted in this reviewer's opinion.

Response

We thank the reviewer for reviewing our revised manuscript.